# CLOVER: Cross-Layer Orthogonal Vectors Pruning

**Fanxu Meng** [1 2] **Pingzhi Tang** [1] **Fan Jiang** [1] **Muhan Zhang** [1 2]

## Abstract

Decoder-only models generate tokens autoregressively by caching key/value vectors, but as the cache grows, inference becomes memory-bounded. To address this challenge, we introduce CLOVER (Cross-Layer Orthogonal Vectors) pruning, a novel approach that treats pairs of components of the attention mechanism as low-rank decompositions. CLOVER applies Singular Value Decomposition (SVD) to the Q-K and V-O pairs within each attention head. The resulting singular values, in turn, guide pruning and further serve as trainable parameters for efficient fine-tuning, ultimately enabling the model to recover its performance to the level before pruning. After pruning and fine-tuning, these values are reintegrated into the model without increasing its parameter count. Visualizations across various models show that CLOVER effectively removes linear redundancies within attention heads, greatly improving pruning efficiency. For example, pruning 70% of the Q-K head dimension in GPT-2 XL results in a perplexity comparable to that of pruning just 8% using vanilla pruning. The combination of CLOVER and TransMLA achieves a speedup of up to 11.1× over LLaMA-2-7B. Our code is available at: https://github.com/GraphPKU/CLOVER

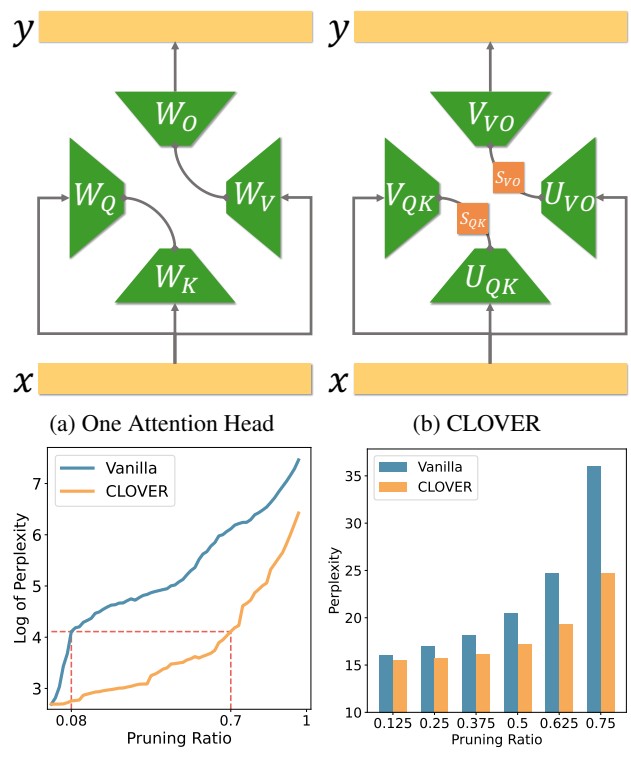

(a) One Attention Head      (b) CLOVER

(c) Pruning without Training      (d) Fine-Tuning Pruned Model

*Figure 1.* (a) We treat the Query-Key and Value-Output layers within a single attention head as a unified structure. (b) Apply SVD to obtain two sets of singular vectors for initializing the Q-K and V-O layers, along with singular values that guide pruning or enable efficient full-rank fine-tuning. (c) This cross-layer orthogonalization strategy allows for higher pruning rates. (d) The pruned model maintains strong performance after fine-tuning.

## 1. Introduction

In recent years, Large Language Models (LLMs) have rapidly evolved into essential tools for productivity (OpenAI, 2024; Anthropic, 2024; Team et al., 2024a). Open-source models (AI@Meta, 2024; Mistral, 2024; Qwen, 2024; Liu et al., 2024b; Team et al., 2024b; Abdin et al., 2024) have also narrowed the performance gap with closed-source models. The success of LLMs is largely attributed to Next Token

Prediction (Radford et al., 2018; Brown et al., 2020), where tokens are predicted sequentially, with attention computed between each token and all preceding ones. To avoid redundant computations, key-value features are cached. However, as model size grows, the overhead of caching becomes substantial, leading to memory and communication bottlenecks. For instance, in the case of a 65B parameter model (Touvron et al., 2023) with 8-bit key-value quantization, storing 512K tokens requires over 86GB of GPU memory, which surpasses the capacity of a single 80GB GPU (Sun et al., 2024).

[1]Institute for Artificial Intelligence, Peking University [2]State Key Laboratory of General Artificial Intelligence, BIGAI. Correspondence to: Muhan Zhang <muhan@pku.edu.cn>.

*Proceedings of the 42$^{nd}$ International Conference on Machine Learning*, Vancouver, Canada. PMLR 267, 2025. Copyright 2025 by the author(s).

To enable efficient training and inference, we introduce CLOVER (Cross-Layer Orthogonal Vectors) pruning, a novel method that orthogonalizes the Query, Key, Value, and Output vectors without generating additional transformation matrices. As shown in Figure 1a, we treat the $Q$-$K$ and $V$-$O$ pairs in each attention head as a low-rank decomposition of $W_{QK}$ and $W_{VO}$. By crossing these layers and performing SVD on $W_{QK}$ and $W_{VO}$, the Query, Key, Value, and Output vectors become orthogonal within each attention head. Figure 1b illustrates how the resulting singular values can guide pruning or serve as trainable parameters for efficient fine-tuning. After pruning or fine-tuning, these values can be reintegrated into the model without increasing its parameter count. Notably, previous methods, such as SVFT (Lingam et al., 2024), obtain orthogonal vectors by directly performing orthogonal decomposition on each projection matrix, which can lead to the introduction of a large number of additional parameters during fine-tuning. In contrast, CLOVER jointly decomposes $Q$-$K$ and $V$-$O$ pairsas transformation matrices for each other. CLOVER only generates a small set of singular values to guide pruning and fine-tuning, which can be merged back into the model without increasing inference costs.

**By orthogonalizing the vectors, we eliminate linear redundancy.** Attention heads contain numerous non-zero norm vectors. Directly pruning these vectors would degrade performance, but orthogonalizing them allows us to represent the entire attention head's space using a small set of orthonormal basis. The remaining vectors are nearly zero, making them safe to prune. As shown in Figure 1c, pruning 70% of the total budget in the query-key pair using CLOVER—where the pruning ratio can vary across different attention layers—yields a perplexity comparable to that of vanilla pruning, which removes only 8% of the vectors. We summarize the contribution of our paper as follows:

- We treat the Q-K and V-O pairs in each attention head as low-rank approximations of $W_{QK}$ and $W_{VO}$. By performing SVD, we orthogonalize the attention head without adding extra transformation matrices.

- This orthogonalization not only reduces linear redundancy, but also is compatible with any other pruning method, thereby allowing for higher pruning ratios. Pruning 46.42% of the vectors in Whisper's attention head—OpenAI's speech-to-text model(Radford et al., 2023)—preserves performance, without the need for additional training.

- CLOVER enables parameter-efficient fine-tuning, surpassing SOTA methods such as LoRA, DoRA, HiRA, and PiSSA on eight commonsense reasoning tasks across LLaMA-2-7B, and LLaMA-3-8B. Additional analyses further highlight its advantages.

## 2. Related Work

**LLM Compression** To alleviate the memory pressure of KV caches in long-context models, researchers have explored several complementary directions: dynamic token pruning keeps only the past tokens whose attention scores materially influence future predictions, discarding the rest at inference time (Fu et al., 2024; Jo & Shin, 2024; Li et al., 2024b); rank compression of keys and values factorizes or groups the K/V tensors so that a lower-rank set of basis vectors plus coefficients can replicate the original attention, cutting the cache size almost linearly with the reduced rank (Shazeer, 2019; Ainslie et al., 2023; Liu et al., 2024a; Yu et al., 2024); head- or dimension-level pruning statistically identifies and removes low-impact attention heads or sub-dimensions, slimming each token's stored representation (Ashkboos et al., 2024; Xia et al., 2023; Sun et al., 2023); cross-layer KV sharing reuses one KV table across multiple transformer layers, turning layer-wise memory growth into a constant factor (Sun et al., 2024; Brandon et al., 2024; Liu et al., 2024c; Zuhri et al., 2024); and quantization encodes KV weights and activations in INT4–INT8 (or fewer bits), shrinking the cache size without altering the computational graph (Frantar et al., 2022; Dettmers et al., 2022; Xiao et al., 2023; Liu et al., 2024e; Hooper et al., 2024). Although these compression methods can reduce the model size and may ultimately achieve inference speedup, dropping active dimensions inevitably hurts accuracy (Ma et al., 2023); fortunately, parameter-efficient fine-tuning (PEFT) methods—e.g., LoRA or adapters inserted after pruning—restore most of the lost quality with just 0.1–1 % extra trainable parameters (Guo et al., 2023).

**Parameter Efficient Fine-Tuning.** Low Rank Adaption (LoRA) is widely used due to its simplicity and effectiveness, with recent works enhancing it further (Zhang et al., 2023; Zi et al., 2023; Liu et al., 2024d; Zhao et al., 2024; Jiang et al., 2024). PiSSA (Meng et al., 2024) improves convergence speed by initializing adapters with principal singular values and vectors, also reducing quantization error (Wang et al., 2024a;b; Li et al., 2024a). However, PiSSA is limited by its use of a fixed set of orthogonal bases. SVFT (Lingam et al., 2024) directly applies Singular Value Decomposition (SVD) to the original matrix, but this increases the number of parameters, raising computational overhead and reducing efficiency. The CLOVER method addresses these issues by treating the Query-Key pairs in each attention head as low-rank matrices. Using orthogonal decomposition, CLOVER eliminates the need for additional transformation matrices. Instead, it leverages a small set of singular values to linearly combine orthogonal vectors, making the approach more parameter-efficient. After fine-tuning, the adapter can be smoothly reintegrated into the original matrix structure.

# 3. CLOVER: Cross-Layer Orthogonal Vectors

Below is a step-by-step breakdown of the CLOVER method, illustrating how it performs orthogonalization of the Query, Key, Value, and Output layers in Multi-Head Attention, how orthogonal initialization helps improve pruning rates, and how the singular value matrices obtained from orthogonal decomposition can be used for efficient parameter fine-tuning.

We begin by using the computation of the $Q$-$K$ pair as a representative example, which is then generalized to the $V$-$O$ pair.

**Multi-Head Self-Attention**   In a multi-head self-attention mechanism with $H$ heads, each head $h \in \{1, \ldots, H\}$ computes an attention score as:

$$\text{attn}(Q_h, K_h) = \text{softmax}\left(\frac{Q_h K_h^\top}{\sqrt{d}}\right),$$

where $d$ is the dimension of each head, $Q_h, K_h \in \mathbb{R}^{n \times d}$ are the query and key representations for head $h$.

Specifically, the queries and keys for head $h$ are obtained by multiplying the input matrix $X \in \mathbb{R}^{n \times D}$ ($n$ is the sequence length, $D$ is the hidden dimension) with the corresponding "slice" of projection matrices $W_Q, W_K \in \mathbb{R}^{D \times H \times d}$, respectively:

$$Q_h = X\, W_Q^{[:,h,:]}, \quad K_h = X\, W_K^{[:,h,:]}.$$

**Cross-Layer Merging**   Substituting the expressions for $Q_h$ and $K_h$ into the product $Q_h K_h^\top$, we have:

$$Q_h K_h^\top = X\, W_Q^{[:,h,:]} \left(W_K^{[:,h,:]}\right)^\top X^\top.$$

Notice that the original weights $W_Q^{[:,h,:]}$ and $W_K^{[:,h,:]}$ are each in $\mathbb{R}^{D \times d}$. When multiplied together, the resulting matrix $W_{QK}^h = W_Q^{[:,h,:]} \left(W_K^{[:,h,:]}\right)^\top$ has dimensions $D \times D$. Since $d \ll D$, directly using $W_{QK}^h$ in computations—or storing it as trainable parameters—would be highly inefficient, limiting the applicability of such parameter merging.

**Cross-Layer Orthogonal Decomposition**   To mitigate the large size of $W_{QK}^h$, we factorize it via SVD:

$$W_{QK}^h = U_{QK}^h\, S_{QK}^h\, V_{QK}^h,$$

where $U_{QK}^h$ and $V_{QK}^h$ are $D \times D$ orthogonal matrices, $S_{QK}^h$ is a $D \times D$ diagonal matrix of singular values.

Since $W_Q^{[:,h,:]}$ and $W_K^{[:,h,:]}$ each have dimensions $\mathbb{R}^{D \times d}$, the rank of $W_{QK}^h$ is at most $d$. Thus, the number of **nonzero**

singular values in $S_{QK}^h$ are **at most** $d$. We can truncate the SVD to retain only the top-$r$ singular values without any loss of information:

$$W_{QK}^h = U_{QK}^h[:, :r]\, S_{QK}^h[:r, :r]\, V_{QK}^h[:r, :],$$

where $r \le d$.

The process can be easily applied to $W_V$ and $W_O$, as detailed in Appendix D.4.

**CLOVER for Pruning**   After performing SVD, we can rewrite the weight matrix $W_{QK}^h$ as follows:

$$W_{QK}^h = \underbrace{U_{QK}^h[:, :r]\, S_{QK}^h[:r, :r]}_{\widetilde{W}_Q^h}\, \underbrace{\left(V_{QK}^h[:, :r]\right)}_{\widetilde{W}_K^h}^\top.$$

Instead of storing the full matrices $W_Q^h$ and $W_K^h \in \mathbb{R}^{D \times d}$, we store the smaller factors $\widetilde{W}_Q^h$ and $\widetilde{W}_K^h \in \mathbb{R}^{D \times r}$, which are significantly smaller than the original matrix since $r \le d \ll D$. This leads to a reduction in both memory usage and computational cost. Additionally, we can further **prune** small nonzero singular values (and their corresponding singular vectors) that fall below a chosen threshold, further reducing the parameter count and computational overhead.

**CLOVER for Fine-Tuning**   CLOVER can be used not only for pruning, but also for parameter-efficient fine-tuning. We freeze the matrices $U_{QK}^h[:, :r]$ and $V_{QK}^h[:, :r]$, and only fine-tune the singular values $S_{QK}^h[:r, :r]$.

In contrast to SVFT, which factorizes the original weight matrices $W_Q, W_K, W_V, W_O \in \mathbb{R}^{D \times H \times d}$ individually, CLOVER factorizes the merged weights $W_{QK}^h$ and $W_{OV}^h$ within each attention head. As a result, the tunable matrix $S_{QK}$ has a size bounded by $\mathbb{R}^{H \times d \times d}$ (considering all heads). In comparison, SVFT requires factorizing large matrices each into three components ($U, S, V \in \mathbb{R}^{D \times D}$), leading to a significant increase in parameter count and computational overhead, even with sparse updates for the singular values $S$.

For example, consider the LLaMA 2-7B model with $H = 32$ attention heads and a head dimension of $d = 128$. By factorizing each head separately, the largest size for $S_{QK}$ is $\mathcal{O}(32 \times 128 \times 128)$, which is significantly smaller than factorizing a $\mathbb{R}^{4096 \times 4096}$ matrix. This makes CLOVER's parameter efficiency comparable to that of a LoRA configuration with rank 32, as shown in Appendix B, but with additional potential for pruning.

*Table 1.* Comparison of CLOVER with SliceGPT and TransMLA on pruning DeepSeek-V2-Lite and LLaMA-2-7B separately, and evaluation of their fine-tuned performance across six benchmarks.

| Model | Hidden Size | Head Dim | Avg. | MMLU | ARC | PIQA | HS | OBQA | WG |
|---|---|---|---|---|---|---|---|---|---|
| DeepSeek V2 Lite | – | – | 61.54 | 43.29 | 60.39 | 79.92 | 74.51 | 45.40 | 65.75 |
| *- SliceGPT* | -6.25% | – | 57.30 | 38.40 | 55.95 | 77.20 | 68.67 | 41.20 | 62.35 |
| | -12.50% | – | 53.51 | 35.24 | 51.97 | 74.27 | 62.08 | 37.80 | 59.67 |
| *- CLOVER* | – | -25% | 59.84 | 41.16 | 57.56 | 79.27 | 72.61 | 44.60 | 63.85 |
| | – | -50% | 57.25 | 38.96 | 55.27 | 78.02 | 69.63 | 41.40 | 60.22 |

| Model | KV Cache | Head Dim | Avg. | MMLU | ARC | PIQA | HS | OBQA | WG |
|---|---|---|---|---|---|---|---|---|---|
| LLaMA-2-7B | – | – | 59.85 | 41.43 | 59.24 | 78.40 | 73.29 | 41.80 | 64.96 |
| *- TransMLA* | -68.75% | – | 59.82 | 40.87 | 59.18 | 77.91 | 71.82 | 45.20 | 63.93 |
| | -87.50% | – | 59.36 | 40.77 | 58.84 | 78.18 | 71.28 | 43.60 | 63.46 |
| | -92.97% | – | 58.68 | 40.82 | 59.72 | 76.55 | 69.97 | 43.60 | 61.40 |
| *- CLOVER* | -68.75% | -50% | 59.40 | 40.91 | 58.97 | 78.35 | 71.32 | 43.40 | 63.46 |
| | -87.50% | -50% | 59.28 | 40.46 | 59.12 | 77.48 | 70.62 | 44.60 | 63.38 |
| | -92.97% | -50% | 59.13 | 40.69 | 60.03 | 77.09 | 69.65 | 45.20 | 62.12 |

*Table 2.* Comparison of latency metrics for different method.

| Model | Prefilling (ms) | Generation (ms/token) |
|---|---|---|
| DeepSeek | 195.12 | 40.11 |
| SliceGPT | 191.91 | 40.32 |
| CLOVER | 177.02 | 31.00 |

## 4. Experiments

In Section 4.1, we compare CLOVER with SliceGPT (Ashkboos et al., 2024) and TransMLA (Meng et al., 2025), which respectively prune DeepSeek-v2-Lite (DeepSeek-AI, 2024) and LLaMA-2-7B (AI@Meta, 2023). In Section 4.2, we visualize how CLOVER removes linear redundancy between vectors, facilitating more efficient pruning. In Section 4.3, we evaluate the acceleration performance of CLOVER. In Section 4.4, we demonstrate CLOVER's ability to perform significant pruning In Section 4.5, we apply CLOVER to orthogonalize the attention heads of the GPT-2-XL model (Radford et al., 2019), to explore the role of CLOVER in both pruning and fine-tuning. In Section 4.6, we conduct fine-tuning experiments on eight commonsense tasks, comparing CLOVER with SOTA PEFT methods.

### 4.1. Comparing CLOVER with Other Methods

Currently, pruning efforts for DeepSeek models are limited. The few existing approaches mainly focus on reducing the number of experts in the MoE module (Gu et al., 2025). However, by orthogonalizing the attention heads in DeepSeek, we observe that significant redundancy also exists within MLA (Figure 2a). Removing this redundancy can substantially reduce the computational overhead during training, pre-filling, and the computation of query represen-

tations in the absorb phase. To compare the effectiveness of CLOVER with other pruning methods, we adapted the SliceGPT (Ashkboos et al., 2024) codebase to support the DeepSeek model architecture. And we applied CLOVER to orthogonally initialize the attention heads and pruned the attention head dimensions based on the magnitude of singular values.

Additionally, we compared pruning for LLaMA-2-7B with TransMLA (Meng et al., 2025), which converts models using Multi-Head Attention (MHA) or Grouped Query Attention (GQA) into MLA-based models, effectively compressing the KV cache. TransMLA can be further combined with CLOVER to prune the dimensionality of attention heads more efficiently. We pruned the K NoPE and V head dimensions in the LLaMA-2-7B model released in their paper, to evaluate CLOVER's effectiveness. For fine-tuning, we followed the TransMLA procedure on a mixed pretraining dataset, as shown in Table 6.

All models were evaluated on six benchmarks: MMLU (Hendrycks et al., 2021), ARC (easy and challenge) (Clark et al., 2018a), PIQA (Bisk et al., 2020a), HellaSwag (HS) (Zellers et al., 2019a), and Winogrande (WG) (Sakaguchi et al., 2021a). These evaluations serve to validate the effectiveness of different pruning strategies.

As shown in Table 1, CLOVER achieves performance comparable to SliceGPT while pruning 50% of the head dimension, compared to SliceGPT's 6.25% pruning. However, as demonstrated in Table 2, CLOVER delivers a 1.25× speedup, whereas SliceGPT provides no acceleration. Furthermore, building on TransMLA, an additional 50% pruning of the head dimension still allows the model to recover its performance with only a small amount of retraining.

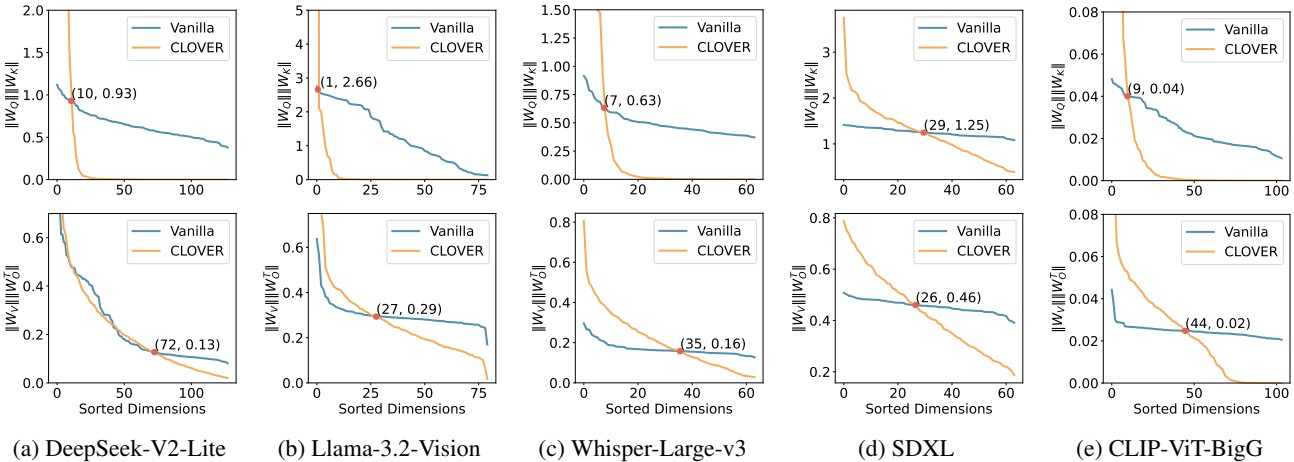

*Figure 2.* CLOVER (orange) uses fewer orthogonal basis vectors than Vanilla Pruning (blue) to span the attention head space. The first row shows the importance of Q-K dimensions, and the second row shows V-O dimensions. After the red dot, CLOVER's importance is lower, and pruning these vectors results in less performance loss.

### 4.2. CLOVER Removal Redundant Vectors

CLOVER achieves a higher pruning ratio due to the significant linear redundancy present in the model. By representing the entire attention head with only a small number of orthogonal vectors, CLOVER effectively removes this redundancy. To illustrate the advantages of CLOVER in eliminating linear redundancy, we apply it to a variety range of models, including the large language model DeepSeek-V2-Lite (DeepSeek-AI, 2024), the multimodal automatic speech recognition and speech translation model Whisper-Large-v3 (Radford et al., 2023), the multimodal instruction-tuned image reasoning generative models LLaMA-3.2-11B-Vision (AI@Meta, 2024), the image encoder CLIP-ViT-bigG (Cherti et al., 2022), and the image generation model Stable Diffusion XL (Podell et al., 2023). We compute the $L_2$ norm for each dimension (equal to singular values) in both the Q-K pair and the V-O pair, sorting the values in descending order within each attention head for better visualization. For comparison, we also perform Vanilla Pruning, which does not utilize CLOVER initialization but instead sorts directly based on the $L_2$ norm.

Figure 2 showcases the first attention head from the first layer of each model. In the first column of the figure, depicting the Q-K norm, we observe that in the original model, the importance of each dimension is relatively balanced (e.g. Figure 2c). This balanced distribution is a result of the linear redundancy, where different directions are intertwined, making it challenging to prune individual directions without negatively affecting the model's performance. However, after applying CLOVER's orthogonal decomposition, only a small number of orthogonal bases on the left side exhibit significantly large norms. These vectors span almost the entire attention head's space, and the remaining

vectors have norms that approach zero, indicating that they are already represented by the dominant singular vectors and can be pruned without loss of performance. Beyond the red intersection point, CLOVER's remaining vectors exhibit consistently lower importance than those in Vanilla Pruning, meaning pruning these vectors results in less performance degradation. This demonstrates why CLOVER enables a higher pruning ratio. A similar trend is observed for the V-O pair, although the model's inherent sparsity is less pronounced than in the Q-K pair, making the effect less noticeable. Still, in most models, pruning half of the vectors has a smaller impact on performance compared to Vanilla Pruning. Notably, in CLIP-ViT-bigG (Figure 2e), a proportion of the vectors already have a norm of zero, allowing for safe pruning.

Beyond the red intersection point, CLOVER's remaining vectors exhibit consistently lower importance than those in Vanilla Pruning, meaning pruning these vectors results in less performance degradation. This demonstrates why CLOVER enables a higher pruning ratio. A similar trend is observed for the V-O pair, although the model's inherent sparsity is less pronounced than in the Q-K pair, making the effect less noticeable. Still, in most models, pruning half of the vectors has a smaller impact on performance compared to Vanilla Pruning. Notably, in CLIP-ViT-bigG (Figure 2e), a proportion of the vectors already have a norm of zero, allowing for safe pruning.

Beyond the red intersection point, CLOVER's remaining vectors exhibit consistently lower importance than those in Vanilla Pruning, meaning pruning these vectors results in less performance degradation. This demonstrates why CLOVER enables a higher pruning ratio. A similar trend is observed for the V-O pair

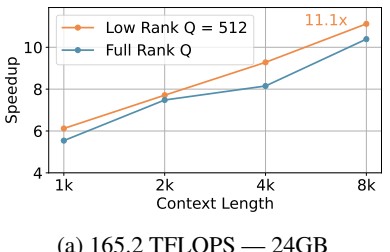
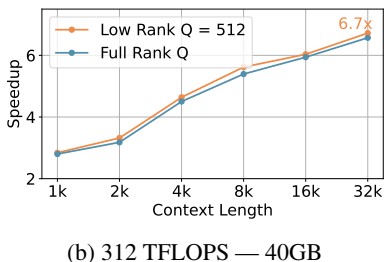
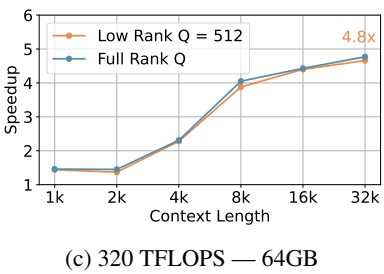

(a) 165.2 TFLOPS — 24GB     (b) 312 TFLOPS — 40GB     (c) 320 TFLOPS — 64GB

*Figure 3.* Inference speedups with CLOVER comparing to the original LLaMA2 7B model on three platform. **Low-rank Q** and **Full-rank Q** indicate whether the query projections were also compressed. **Context length** represents the total sequence length.

### 4.3. Inference Speedup with CLOVER

In Figure 3, we benchmark the inference performance of CLOVER—featuring a 92.97% reduction in the KV cache and a 50% reduction in the Q nope, K nope, and V head dimensions—using the vLLM framework across three GPUs with varying compute capabilities and memory sizes: 165.2 TFLOPS with 24GB memory, 312 TFLOPS with 40GB memory, and 320 TFLOPS with 64GB memory. The figure illustrates the inference speedup of the pruned model relative to the original LLaMA-2-7B. "Low-rank Q" and "Full-rank Q" indicate whether the query projections were also compressed. The context length refers to the total sequence length, which includes both the prompt and generated tokens (with equal lengths for each).

Our experiments demonstrate that CLOVER's inference speedup increases with longer context lengths. As long sequences typically lead to both compute and memory bottlenecks, compressing the KV cache and attention head dimensions helps alleviate these issues, thereby enabling higher speedups. Notably, for an 8K context window on the first hardware platform, the CLOVER-pruned model achieves an impressive 11.1× inference acceleration.

### 4.4. CLOVER for Training-Free Pruning

As demonstrated by the prominent low-rank properties in Figure 2c, we applied pruning to the Whisper-large-v3 model (Radford et al., 2023). We use the official Whisper-large-v3 example (LibriSpeech Long dataset (Gandhi et al., 2023)[1]) to intuitively highlight the effectiveness of CLOVER pruning. For reference, the waveform of this input is shown in Figure 4, and the corresponding target translation script is provided in Appendix C.

After applying CLOVER to orthogonalize the vectors, we pruned vectors with magnitudes close to zero ($\|W_Q\|\|W_K\| \leq 5 \times 10^{-3}$ and $\|W_V\|\|W_O^\top\| \leq 6 \times 10^{-3}$). This pruning achieved ratios of 56.01% and 36.82% for the parameters in $Q$-$K$ Pair and $V$-$O$ Pair, respectively. Re-

---

[1]https://huggingface.co/openai/whisper-large-v3

markably, the model's output remains nearly unchanged, with only one error, which has been highlighted in the text using strikethrough and red for clarity:

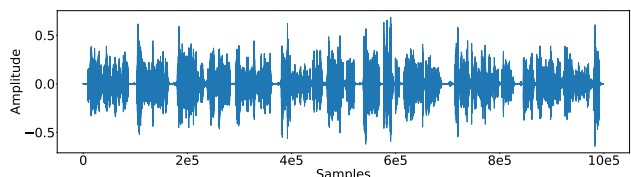

*Figure 4.* An audio waveform from the librispeech dataset.

*Mr. Quilter is the apostle of the middle classes, and we are glad to welcome his gospel. Nor is Mr. Quilter's manner less interesting than his matter. He tells us that at this festive season of the year, with Christmas and roast beef looming before us, similes drawn from eating and its results occur most readily to the mind. He has grave doubts whether Sir Frederick Layton's work is really Greek after all, and can discover in it but little of rocky Ithaca. Linnell's pictures are a sort of Up Guards and Adam paintings, and Mason's exquisite idles are as national as a jingo poem. Mr. Birkett Foster's landscapes smile at one much in the same way that Mr. Carker used to flash his teeth. ~~And~~, and Mr. John Collier gives his sitter a cheerful slap on the back before he says, like a shampooer in a Turkish bath, next man.*

In contrast, vanilla pruning—which forgoes orthogonal initialization and prunes head vectors solely based on their norm—results in the model completely failing to generate valid outputs at the same pruning ratio.

... ... ... ... ... ... ... ... ... ... ... ... ... ... ... ... ... ... ... ... ... ...

This example validates our earlier claim that straightforward pruning of non-zero dimensions can lead to accumulated loss. In contrast, CLOVER effectively eliminates linear redundancy, enabling a significantly higher pruning ratio. When the linear redundancy is sufficiently pronounced, CLOVER can even achieve a high pruning ratio without the need for fine-tuning to recover performance.

*Table 3.* Pruning the attention layers of GPT-2-XL using CLOVER and vanilla pruning at various sparsity levels. We report perplexity on WikiText-2 (lower is better) and evaluate fine-tuning performance on OpenWebText under different token budgets. The base model's perplexity is 14.78. CLOVER**FT** and Vanilla fine-tune the pruned attention layers, while CLOVER**PEFT** fine-tunes the singular value matrices obtained from the decomposition of the QK and VO projections.

| Pruning Ratio | w/o Training Perplexity(↓) | | 66M Tokens Perplexity (↓) | | | 131M Tokens Perplexity (↓) | | |
|---|---|---|---|---|---|---|---|---|
| | Vanilla | CLOVER | Vanilla | CLOVER**FT** | CLOVER**PEFT** | Vanilla | CLOVER**FT** | CLOVER**PEFT** |
| 12.5% | 33.76 | **15.89** | 16.04 | **15.45** | 15.67 | 16.38 | 15.77 | **15.42** |
| 25.0% | 78.36 | **17.45** | 16.93 | **15.70** | 15.89 | 17.07 | 16.05 | **15.75** |
| 37.5% | 159.4 | **20.95** | 18.17 | **16.17** | 16.60 | 18.14 | 16.48 | **16.41** |
| 50.0% | 338.9 | **35.12** | 20.45 | **17.22** | 17.63 | 19.02 | **17.13** | 17.71 |
| 62.5% | 538.5 | **85.25** | 24.65 | **19.32** | 20.64 | 21.44 | **18.40** | 20.39 |
| 75.0% | 708.8 | **187.4** | 36.04 | **24.65** | 29.28 | 27.22 | **20.99** | 28.44 |

## 4.5. Pruning and Fine-Tuning with CLOVER

Model pruning often necessitates fine-tuning to recover from performance degradation. CLOVER supports both pruning and fine-tuning within a unified framework. In this section, we evaluate CLOVER's effectiveness in both aspects. We initialize GPT-2-XL with CLOVER and prune the model by removing vectors corresponding to the singular values with the smallest magnitudes.

To achieve better performance, Figure 1c allows each layer to adopt a different pruning rate. In this case, a fixed proportion of parameters is pruned across the entire model based on the global ranking of singular values (for CLOVER) or $L_2$-norms (for Vanilla pruning). Remarkably, pruning 70% of the parameters using CLOVER yields performance comparable to pruning only 8% with Vanilla pruning, highlighting the effectiveness of CLOVER's orthogonal initialization in facilitating structured pruning.

However, using a uniform pruning rate across all layers—where the same percentage of the smallest singular vectors is pruned per layer—is beneficial for consistent training and inference speed. Therefore, unless otherwise noted, we apply uniform pruning across layers. After fine-tuning, the singular values $S$ are merged into their corresponding $U$ and $V$ matrices. For comparison, we also apply Vanilla pruning, which lacks CLOVER's orthogonalization and instead uses an $L_2$-norm-based criterion.

Following pruning, we evaluate model perplexity on the WikiText-2 dataset (Merity et al., 2016). We then fine-tune the pruned models on the OpenWebText dataset (Gokaslan & Cohen, 2019), using the nanoGPT framework[2]. To minimize disruption to the pretrained model, only the pruned attention layers are fine-tuned, while the MLP, embedding layers, and LM head remain fixed. This setup is referred to as CLOVER**FT** and Vanilla, respectively. In the CLOVER**PEFT** configuration, the singular values $S$ are not immediately

merged into the $U$ and $V$ matrices. Instead, they are retained for parameter-efficient fine-tuning, where only these singular values are updated, and merging is deferred until post-training. PEFT typically converges more slowly than full-parameter finetuning. To accelerate convergence, we increase the learning rate from $6 \times 10^{-4}$ to $6 \times 10^{-3}$ and remove weight decay, while keeping all other hyperparameters consistent with those used in Vanilla and CLOVER**FT**.

As shown in Table 3, CLOVER induces significantly less performance degradation than Vanilla pruning by concentrating functionality into fewer orthogonal bases. For instance, pruning 50% of the parameters without fine-tuning increases CLOVER**FT**'s perplexity by only 1.38×, compared to 21.9× for Vanilla. After fine-tuning, CLOVER**FT** substantially outperforms Vanilla; for example, CLOVER with a 75% pruning rate achieves comparable performance to Vanilla pruning at only 62.5%. Owing to its reduced model disruption, CLOVER**FT** also requires fewer training tokens to restore performance (e.g., perplexity with 66M tokens closely matches that with 131M), whereas Vanilla pruning demands more data, increasing both computational cost and the risk of degradation on out-of-domain tasks.

Moreover, CLOVER**PEFT**, which fine-tunes only the singular values from the SVD decomposition and the attention layer biases, enables performance recovery with minimal resource consumption and parameter updates. At lower pruning rates, CLOVER**PEFT** even surpasses full attention-layer training (CLOVER**FT**). However, at higher pruning rates, performance declines significantly due to the limited number of remaining tunable parameters (e.g., only 0.15% of the original attention-layer parameters are updated).

These results empirically validate the benefits discussed earlier: CLOVER's orthogonal initialization of attention heads enables the representation of the entire attention space using a compact set of orthogonal bases, which is highly advantageous for pruning. Furthermore, the singular value matrix can be seamlessly merged back into the attention head.

---

[2]https://github.com/karpathy/nanoGPT

*Table 4.* Accuracy comparison of LLaMA2-7B, and LLaMA3-8B with various PEFT methods on eight commonsense reasoning datasets. Results of LoRA and DoRA are taken from (Liu et al., 2024d). Results of HiRA are taken from (Huang et al., 2025).

| Model | Method | Params | BoolQ | PIQA | SIQA | Hella Swag | Wino Grande | ARC-e | ARC-c | OBQA | Avg. |
|---|---|---|---|---|---|---|---|---|---|---|---|
| | LoRA | 0.83% | 69.8 | 79.9 | 79.5 | 83.6 | 82.6 | 79.8 | 64.7 | 81.0 | 77.6 |
| | DoRA | 0.84% | 71.8 | 83.7 | 76.0 | 89.1 | 82.6 | 83.7 | 68.2 | 82.4 | 79.7 |
| LLaMA2-7B | HiRA | 0.83% | 71.2 | 83.4 | 79.5 | 88.1 | 84.0 | 86.7 | 73.8 | 84.6 | 81.4 |
| | PiSSA | 0.83% | **75.0** | **87.0** | 81.6 | 95.0 | 86.5 | 88.5 | 75.9 | 86.4 | 84.5 |
| | CLOVER | 0.83% | **75.0** | 86.4 | **82.0** | **95.1** | **87.5** | **89.6** | **76.6** | **89.4** | **85.2** |
| | LoRA | 0.70% | 70.8 | 85.2 | 79.9 | 91.7 | 84.3 | 84.2 | 71.2 | 79.0 | 80.8 |
| | DoRA | 0.71% | 74.6 | 89.3 | 79.9 | 95.5 | 85.6 | 90.5 | 80.4 | 85.8 | 85.2 |
| LLaMA3-8B | HiRA | 0.70% | 75.4 | 89.7 | 81.2 | 95.4 | 87.7 | 93.3 | 82.9 | 88.3 | 86.7 |
| | PiSSA | 0.70% | **77.2** | **90.0** | **82.9** | 96.6 | 88.4 | **93.6** | 82.4 | 87.4 | 87.3 |
| | CLOVER | **0.47%** | 76.4 | 89.3 | 82.1 | **96.9** | **89.9** | **93.6** | **84.5** | **90.6** | **87.9** |

*Table 5.* Comparison of training costs between LoRA and CLOVER on LLaMA-2-7B. We trained on a commonsense dataset for 3 epochs with model_max_length = 1024, per_device_train_batch_size = 2, gradient_accumulation_steps = 2, num_gpus = 4, executed on 4 × 312 TFLOPS–80G GPUs.

| Method | Params | Max Memory | Runtime |
|---|---|---|---|
| LoRA | 0.83% | 110.84 GB | 2:42:37 |
| CLOVER | 0.83% | 104.75 GB | 2:22:47 |

### 4.6. Comparison with PEFT Methods

In this section, we conduct an ablation study to compare the fine-tuning capability of CLOVER against several parameter-efficient fine-tuning (PEFT) methods, including LoRA (Hu et al., 2021), DoRA (Liu et al., 2024d), HiRA (Huang et al., 2025), and PiSSA (Meng et al., 2024). We exclude SVFT (Lingam et al., 2024) from this comparison due to its significant computational overhead. The evaluation spans eight sub-tasks, as detailed in Table 7. All models are fine-tuned on the Commonsense-148k dataset and evaluated on the respective test sets of each sub-task.

For CLOVER, we apply orthogonal decomposition to the Value-Output projection and fine-tune the resulting singular value matrix. Due to the non-linear RoPE (Su et al., 2024) operation between the query and key, we instead decompose the Key layer and fine-tune its transition matrix. Likewise, in the `mlp.up_proj` layer, we treat every 64 consecutive dimensions as a head, apply orthogonal decomposition, and update the corresponding transition matrix.

The number of trainable parameters in LLaMA-2-7B matches those used in LoRA, DoRA, HiRA, and PiSSA, all employing rank-32 updates. For LLaMA-3-8B, we reduce the number of trainable parameters to two-thirds of the amount used in the other models.

The comparison of memory consumption and runtime in Table 5 demonstrate that CLOVER consumes less GPU memory and exhibits shorter training runtime compared to LoRA. We attribute this to CLOVER being applied between two layers, whereas LoRA operates in parallel with the main branch. This enables sequential computation, eliminating the need to retain the input features of the main branch.

LoRA and DoRA results are taken from the DoRA paper, while HiRA results are sourced directly from its original publication. Since PiSSA has not conducted experiments on commonsense reasoning datasets, we include its performance by reproducing the experiments ourselves. For a fair comparison, we adopt the hyperparameters from DoRA and adjust the learning rates accordingly. As shown in Table 8, CLOVER achieves the best performance with a learning rate of 1e−4, which we apply consistently across both LLaMA-2-7B and LLaMA-3-8B. PiSSA performs best with a learning rate of 2e−5, as reported in its original paper; all other hyperparameters remain unchanged. Due to the stable training behavior observed in both PiSSA and CLOVER, we omit the validation procedure used in DoRA—where the best-performing model is selected every 80 iterations based on the validation set. Instead, we train for the full 3 epochs and use the final model checkpoint for testing.

Table 4 demonstrates that CLOVER consistently outperforms all other methods across all models and tasks. Specifically, on LLaMA-2-7B, CLOVER surpasses LoRA, DoRA, HiRA, and PiSSA by 7.6%, 5.5%, 3.8%, and 0.7%. Even on LLaMA-3-8B, with fewer trainable parameters, CLOVER outperforms by 7.1%, 2.7%, 1.2%, and 0.6%. CLOVER leads in most sub-tasks and ranks second in a few.

These experiments demonstrate that CLOVER possesses strong fine-tuning capabilities, making it effective for recovering performance degradation caused by pruning. Additional analysis is provided in Appendix D.

## 5. Conclusion and Limitations

In this paper, we introduce Cross-Layer Orthogonal Vectors (CLOVER), a method that orthogonalizes vectors within attention heads without requiring additional transformation matrices. This orthogonalization process condenses effective parameters into fewer vectors, improving the pruning ratio. By fine-tuning the singular values obtained through orthogonalization, CLOVER learns linear combinations of orthogonal bases, enabling full-rank updates. When applied to prune 50% of the attention head parameters in GPT-2XL, CLOVER results in a perplexity that is just one-tenth of that achieved by standard pruning methods. For Whisper-Large-v3, CLOVER removes 46.42% of the parameters without fine-tuning, while preserving model performance. Furthermore, when used for fine-tuning, CLOVER outperforms state-of-the-art methods such as LoRA, DoRA, HiRA, and PiSSA, achieving superior results with equal or fewer trainable parameters. We also demonstrate how CLOVER removes linear redundancy to facilitate pruning and discuss the necessity of fine-tuning across all orthogonal bases. Visual comparisons of models fine-tuned with different methods further illustrate its effectiveness.

Despite its advantages, CLOVER has some limitations. When nonlinear operations are present between Q-K or V-O pairs (such as with the widely-used RoPE (Su et al., 2024)), cross-layer orthogonalization is not feasible. In these cases, we instead perform head-wise orthogonalization within the Key layer during fine-tuning. Fortunately, CLOVER Fine-Tuning can apply intra-layer attention head orthogonalization, while CLOVER Pruning remains applicable to many popular models, including DeepSeek (DeepSeek-AI, 2024; Liu et al., 2024b)(which uses Decoupled RoPE), ViT and SDXL (which use absolute positional encoding), and BLOOM (Workshop et al., 2022) (which employs Alibi relative positional encoding (Press et al., 2021)). Additionally, as a newly proposed method, our current evaluation focuses primarily on basic pruning tasks and does not include comparisons with other state-of-the-art pruning techniques. However, because CLOVER does not alter the model structure and only updates the initialization method, it can be combined with existing pruning methods to further enhance their effectiveness.

As a novel technique, CLOVER holds considerable promise for future applications. For instance, it could be combined with quantization methods to eliminate outliers, guide pruning and fine-tuning based on data feature directions, or even inspire new model architectures.

## Acknowledgement

This work is supported by the National Key R&D Program of China (2022ZD0160300), National Natural Science Foundation of China (62276003), and Kunpeng&Ascend Center of Excellence, Peking University.

## Impact Statement

This paper proposes a cross-layer orthogonal initialization method to guide model pruning and efficient fine-tuning, offering valuable insights for the application and development of large models. Both application directions aim to reduce training and inference costs, lower computational overhead, decrease power consumption, and minimize carbon emissions.

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

## A. Dataset and Hyper-Parameters for Table 1

Following the experimental setups of TransMLA (Meng et al., 2025), we fine-tune our models using the prtraining corpus from SmolLM (Ben Allal et al., 2024). The dataset comprises FineWeb-Edu-Dedup (Lozhkov et al., 2024a), Cosmopedia-v2 — a synthetic dataset generated by Mixtral (Wu et al., 2024), Python-Edu from StarCoder (Lozhkov et al., 2024b), Open-Web-Math (Paster et al., 2023), and data from StackOverflow (Stack Overflow, 2025).

*Table 6.* Composition of the training dataset.

| Dataset | Sampling Weight |
|---|---|
| fineweb-edu-dedup | 0.70 |
| cosmopedia-v2 | 0.15 |
| python-edu | 0.06 |
| open-web-math | 0.08 |
| stackoverflow | 0.01 |

To ensure a fair comparison, we replicate the dataset mixing ratios used in the TransMLA setup, as shown in Table 6, to maintain experimental consistency. DeepSeek-V2-Lite is trained on 1B tokens for both SliceGPT and CLOVER using SliceGPT's hyperparameters. For LLaMA-2-7B, we apply CLOVER on top of the checkpoint released by TransMLA, which pruned 93% of the KV cache and was trained on 6B tokens. Additionally, we further train the model with 1.5B tokens for various pruning ratios using TransMLA's hyperparameters.

## B. Dataset and Hyper-Parameters for Table 4

The commonsense reasoning tasks consist of 8 subtasks, each with predefined training and testing sets, as described by LLM-Adapters (Hu et al., 2023). The following table lists the details of each sub-dataset.

*Table 7.* Details of datasets for commonsense reasoning tasks.

| Dataset | Train | Test | About |
|---|---|---|---|
| BoolQ (Clark et al., 2019) | 9,427 | 3,270 | Naturally occurring yes/no questions from unconstrained settings. |
| PIQA (Bisk et al., 2020b) | 16,113 | 1,838 | Questions with two solutions requiring physical commonsense. |
| SIQA (Sap et al., 2019) | 33,410 | 1,954 | Reasoning about actions and social implications. |
| HellaSwag (Zellers et al., 2019b) | 39,905 | 10,042 | Commonsense NLI questions with context and endings. |
| WinoGrande (Sakaguchi et al., 2021b) | **40,398** | 1,267 | Fill-in-the-blank task with binary options. |
| ARC-e (Clark et al., 2018b) | **2,251** | 2,376 | Grade-school multiple-choice science questions in Easy sets. |
| ARC-c (Clark et al., 2018b) | **1,119** | 1,172 | Grade-school multiple-choice science questions in Challenge sets. |
| OBQA (Mihaylov et al., 2018) | 4,957 | 500 | Questions requiring multi-step reasoning and commonsense knowledge. |

For WinoGrande, the original dataset includes multiple partitions: [xs, s, m, l, xl, debiased]. While LLM-Adapters simply concatenated all these partitions, note that the "xl" partition actually includes all others, leading to extensive data duplication. After removing duplicates, the training data is reduced from 63.2K to 40.4K instances.

Additionally, in the LLM-Adapters paper, the training set sizes of ARC_Challenge and ARC_Easy were reversed by mistake; here, we correct that error.

The results for LoRA and DoRA presented in Table 4 are directly taken from the original DoRA paper, where the hyperparameters are carefully tuned. Similarly, the results for HiRA are cited from its original publication. In contrast, we introduce new experimental results for PiSSA and CLOVER, both of which are optimized with the best learning rates (Table 8) and aligned hyperparameters. Specifically, PiSSA achieves optimal performance at a learning rate of 2e-5, while CLOVER performs best at 1e-4, as shown in the table below:

Table 9 presents a comparison of hyperparameters for different fine-tuning methods on commonsense tasks. The target model remains the same for LoRA, DoRA, HiRA, and PiSSA. However, DoRA introduces an additional magnitude module, leading to a slightly higher parameter count. In a single layer of LoRA, the trainable parameters are as follows:

| Method | Learning Rate | Acc |
|--------|---------------|-----|
| | 1e-4 | 80.1 |
| | 5e-5 | 82.9 |
| PiSSA | 3e-5 | 84.1 |
| | 2e-5 | 84.5 |
| | 1e-5 | 83.6 |
| | 5e-4 | 79.0 |
| | 2e-4 | 83.9 |
| CLOVER | 1e-4 | 85.2 |
| | 5e-5 | 84.3 |
| | 2e-5 | 82.8 |

*Table 8.* Learning rate searching.

In LoRA, the trainable parameters are:

$$Q = 4096 \times 32 + 4096 \times 32$$
$$K = 4096 \times 32 + 4096 \times 32$$
$$V = 4096 \times 32 + 4096 \times 32$$
$$\text{Up} = 4096 \times 32 + 11008 \times 32$$
$$\text{Down} = 4096 \times 32 + 11008 \times 32$$

The total sum is 1,753,088.

In CLOVER, the trainable parameters are:

$$QK = 32 \times 128 \times 128$$
$$VO = 32 \times 128 \times 128$$
$$UD = 172 \times 64 \times 64$$

The total sum is also 1,753,088.

Since CLOVER inserts trainable parameters across layers, we use the Q-K pair notation to represent its target model. When CLOVER updates parameters within an attention head, the number of trainable parameters matches exactly that of LoRA at rank 32. To adjust the number of learnable parameters, CLOVER can either span multiple heads or split a single head into multiple blocks. Both PiSSA and CLOVER exhibit stable training performance. Therefore, instead of validating every 80 steps, we omit frequent validation, improving training efficiency.

*Table 9.* Detailed Training Hyperparameters. Q-K,V-O, U-D means CLOVER update pair of orthogonal vectors.

| Method | Target | Evaluation steps | LR | Scheduler | Batch size | Warmup Steps | Epochs |
|--------|--------|------------------|-----|-----------|------------|--------------|--------|
| LoRA | Q,K,V,U,D | 80 | 3e-4 | Linear | 16 | 100 | 3 |
| DoRA | Q,K,V,U,D | 80 | 2e-4 | Linear | 16 | 100 | 3 |
| HiRA | Q,K,V,U,D | 80 | 1e-4/2e-4 | Linear | 32 | 100 | 3 |
| PiSSA | Q,K,V,U,D | – | 2e-5 | Linear | 16 | 100 | 3 |
| CLOVER | Q-K,V-O, U-D | – | 1e-4 | Linear | 16 | 100 | 3 |

## C. LibriSpeech Long dataset target transcript

Below is the reference text of the LibriSpeech Long dataset for comparison.

*Mr. Quilter is the apostle of the middle classes, and we are glad to welcome his gospel. Nor is Mr. Quilter's manner less interesting than his matter. He tells us that at this festive season of the year, with Christmas and roast beef looming before us, similes drawn from eating and its results occur most readily to the mind. He has grave doubts whether Sir Frederick Layton's work is really Greek after all, and can discover in it but little of rocky Ithaca. Linnell's pictures are a sort of Up Guards and Adam paintings, and Mason's exquisite idles are as national as a jingo poem. Mr. Birkett Foster's landscapes smile at one much in the same way that Mr. Carker used to flash his teeth, and Mr. John Collier gives his sitter a cheerful slap on the back before he says, like a shampooer in a Turkish bath, next man.*

In fact, with Vanilla Pruning ratios of just 22.31% and 6.69% for $W_Q$-$W_K$ and $W_V$-$W_O$, respectively, the model's output is already significantly degraded.

*Mr. Colter is the personal of the classes, and we are glad to welcome his gospel. Nor is Mr. Colter's manner less interesting than his manner. He tells us that at this festive season of the year, with Christmas and roast beef looming before us, similarly he is drawn from eating and its results occur most readily to the mind. He is very dull, so very frequently, and is very Greek after all, and can discover in it but little of Rocky Ithaca. The Nell's pictures are sort of up-guard to Adam's paintings, and Mason's exquisite idylls are as national as a jingle poem. Mr. Burke and Foster's landscapes smile at one much in the same way as Mr. Parker, Mr. Flash is tits. And Mr. John Collier gives his sitter a cheerful slap on the back before he says like a shampoo and a Turkish bath, Next man.*

## D. Further Analysis of CLOVER's Fine-Tuning Capability

### D.1. Necessity of Full-Direction Fine-Tuning

Besides pruning with a large ratio, CLOVER is capable of learning linear combinations of all orthogonal vectors within each attention head. This capability allows CLOVER to resemble full-parameter fine-tuning more closely. To highlight the advantages of updating all orthogonal bases, we randomly sampled 16 instances from the Commonsense dataset, fed them into the model, and performed SVD to the model. We then recorded the projection magnitudes of input features along all orthogonal directions. Figure 5 visualizes the results for the middle layer, revealing the following insights:

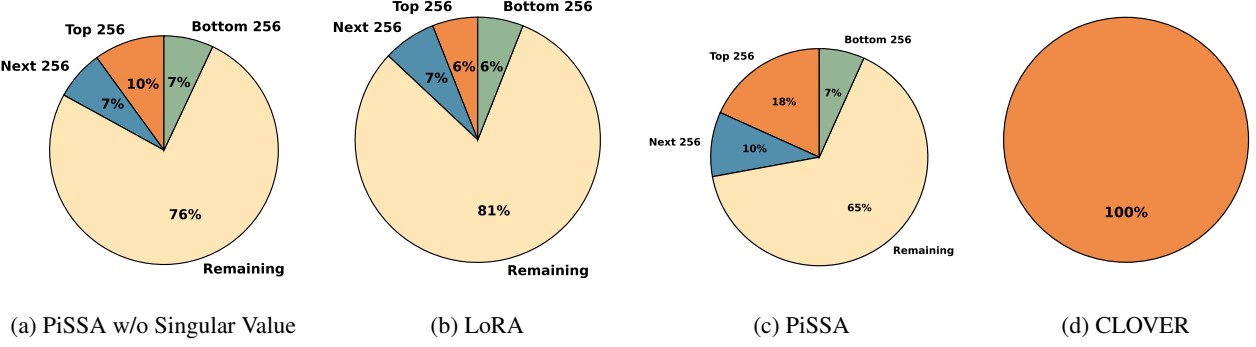

(a) PiSSA w/o Singular Value  (b) LoRA  (c) PiSSA  (d) CLOVER

*Figure 5.* Proportion of data projections across different components in random directions (LoRA) versus orthogonal directions (PiSSA), as well as all orthogonal directions (CLOVER).

1) Without accounting for the scaling effect of singular values, the projection magnitude along the principal singular vector consistently exceeds that in other directions. This observation supports PiSSA's approach, which updates based on the principal singular values and vectors, leading to improved training performance. In contrast, LoRA projects in random directions, resulting in uniform projection magnitudes across all directions.

2) The singular values in the original model reflect the importance of each direction in the pretraining task. The model amplifies the components along directions with larger singular values and suppresses those along smaller singular values. Therefore, it is crucial to consider the scaling effect of singular values. As shown in Figure 5c, the projection magnitude along the principal singular vector direction increases to 18%.

3) While more data projections align with the principal singular vector at higher ranks, 82% of the feature components are still projected onto other directions. In extreme cases, if a task is entirely orthogonal to the vectors used by PiSSA, training

on such a task may result in zero gradients, thereby limiting its learning capacity. Under the same rank constraint, 94% of the feature components in LoRA are projected outside the LoRA adapter, making it more susceptible to the zero-gradient problem.

Since CLOVER updates across all orthogonal directions, as shown in Figure 5d it effectively mitigates this issue. Consequently, CLOVER outperforms both LoRA and PiSSA in multi-task learning, even when using the same or fewer learnable parameters (Section 4.6).

### D.2. Visualizing Rank Updates

To demonstrate CLOVER achieves full-rank updates, we multiply the updated singular values with their corresponding singular vectors and perform SVD on the base model ($S_{QK}$ applied to the Key layer, $S_{VO}$ to the Value layer, and $S_{UD}$ to the Up layer). We take LoRA, and Full Fine-tuning for comparing. Figure 6 shows the singular value of the middle layer in LLaMA-2-7B, revealing that CLOVER and Full Fine-tuning achieve full-rank updates, while LoRA is constrained by its low-rank design.

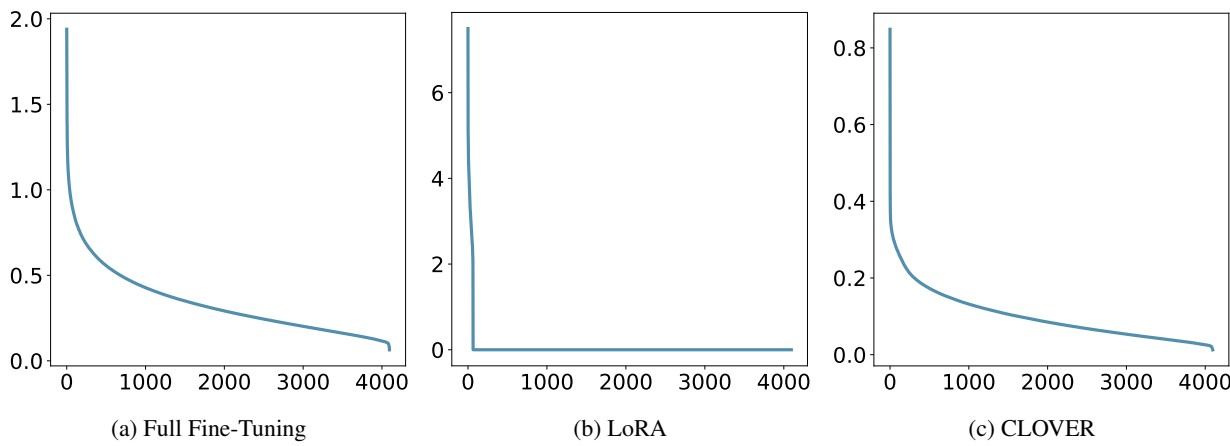

(a) Full Fine-Tuning                    (b) LoRA                    (c) CLOVER

*Figure 6.* $\Delta W$ is low rank in LoRA, while full rank for Full-Fine-Tuning and CLOVER.

### D.3. CLOVER Avoids Intrusive Dimensions

Recent research (Shuttleworth et al., 2024) has highlighted an issue with LoRA, referred to as the "intrusive dimensions" phenomenon. As illustrated in Figure 7b, LoRA introduces new random directions into the model, which possess large magnitudes and thus precede all the original singular vectors. The study suggests that these "intrusive dimensions" can degrade the model's performance, exacerbating catastrophic forgetting during continual learning with LoRA. In contrast, CLOVER addresses this issue by fixing all orthogonal bases and updating only the vector combinations. As a result, the changes introduced by CLOVER fine-tuning closely resemble those generated by full parameter fine-tuning, as shown in Figure 7a and Figure 7c.

### D.4. Cross Layer Orthogonal Vectors in Value and Output layers

In the main text, we only presented the orthogonalization process for the Q-K pair. Here, we provide the method for orthogonalizing the V-O pair. Additionally, for up-down layers, the output dimension of the Up layer can be reshaped into block number × block size, followed by performing orthogonal decomposition within each block.

$$Y = \text{attn}(Q_h, K_h)VW_O, \quad V = XW_V \in \mathbb{R}^{b \times h \times n \times d} \tag{1}$$

$$= \text{attn}(Q_h, K_h)XW_VW_O, \quad W_VW_O = W_{VO} = USV \in \mathbb{R}^{h \times D \times D} \tag{2}$$

$$= \text{attn}(Q_h, K_h)XUSV, \quad S_{[:,r_{vo}:,r_{vo}:]} = S_{VO} \in \mathbb{R}^{h \times r_{vo} \times r_{vo}} = 0, r_{vo} \le d. \tag{3}$$

$$= \text{attn}(Q_h, K_h)XU_{VO}S_{VO}V_{VO}, \quad U_{VO} \in \mathbb{R}^{D \times h \times r_{vo}}, V_{VO} \in \mathbb{R}^{h \times r_{vo} \times D}. \tag{4}$$

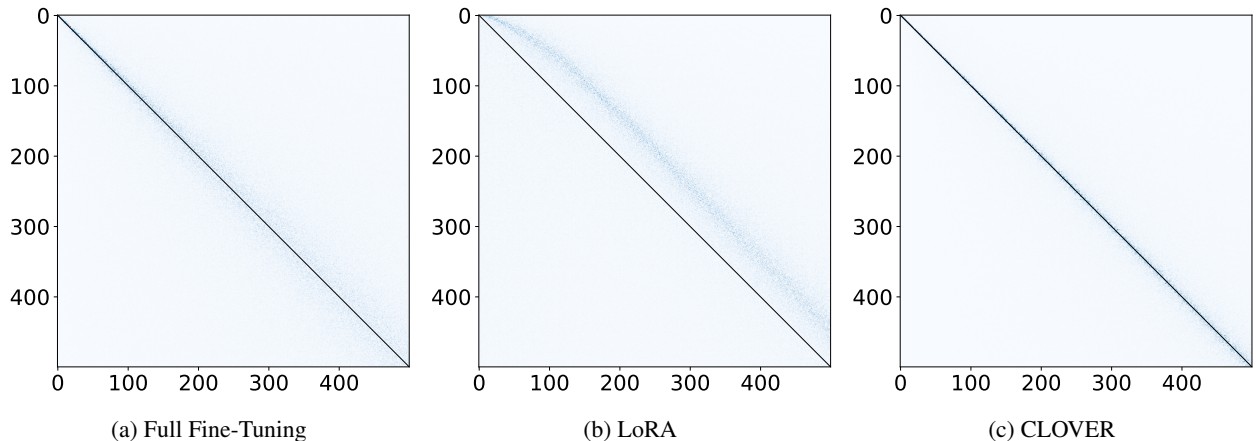

(a) Full Fine-Tuning  (b) LoRA  (c) CLOVER

*Figure 7.* Intruder dimensions phenomenal in LoRA, which does not exist in Full Fine-Tuning and CLOVER.

Through this series of transformations, $W_V$ and $W_O$ can be equivalently replaced by orthogonal vectors $U_{VO}$ and $V_{VO}$, along with the diagonal matrix $S_{VO}$. Since $r_{vo} \leq d$, the singular zero values and their corresponding singular vectors can be safely pruned. After guided pruning, $S_{VO}$ can be merged into $U_{VO}$ and $V_{VO}$, resulting in no additional computational overhead.

## E. Visualizing more attention heads

In Section 4.2, we only presented the first attention head in the first layer. Here, we provide a broader view by showcasing more attention heads. Figure 8 illustrates the $L_2$ norm of all Q-K heads in the first, middle, and last layers of Whisper-Large-v3. Figure 9 shows the $L_2$ norm of all Q-K heads in the first, middle, and last layers of ViT-bigG.

From these figures, we can observe that CLOVER consistently represents the entire attention head with fewer orthogonal bases across all layers and all attention heads. This property forms the foundation of CLOVER's effectiveness in enhancing pruning.

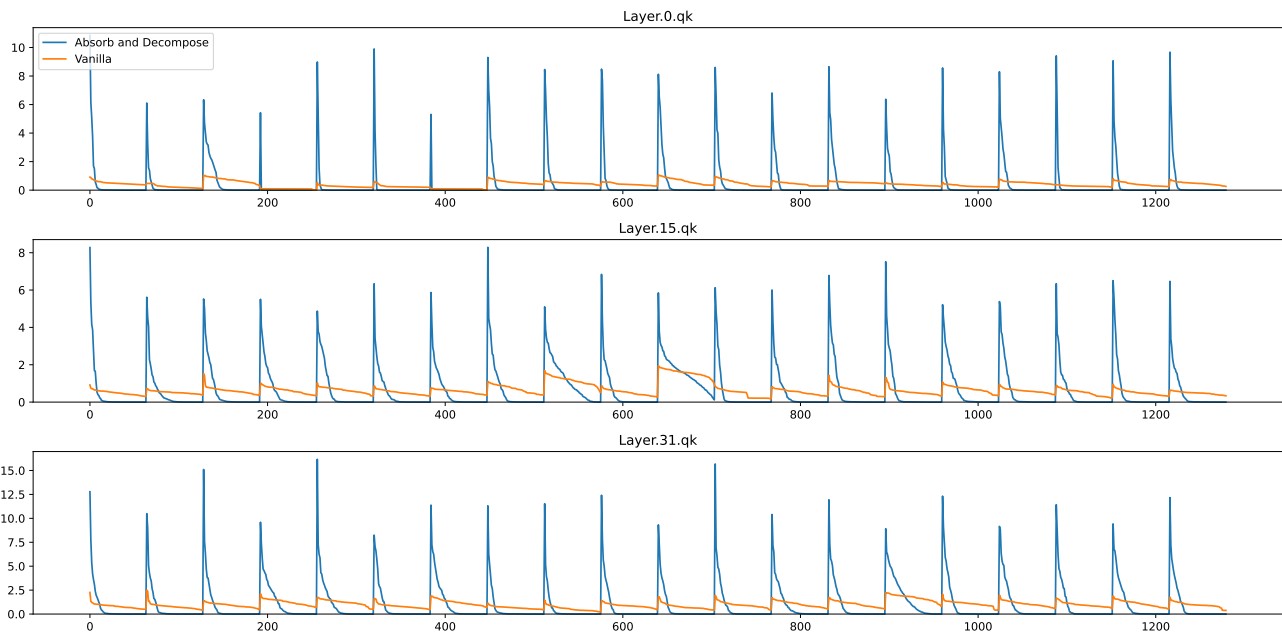

Figure 8. The $L_2$-norm for the 0-th, 15-th, and 31-st attention layers in the Whisper-large-v3 encoder. The blue line represents the results after redundancy removal using the CLOVER method, while the orange line depicts the $L_2$-norm directly computed for each dimension.

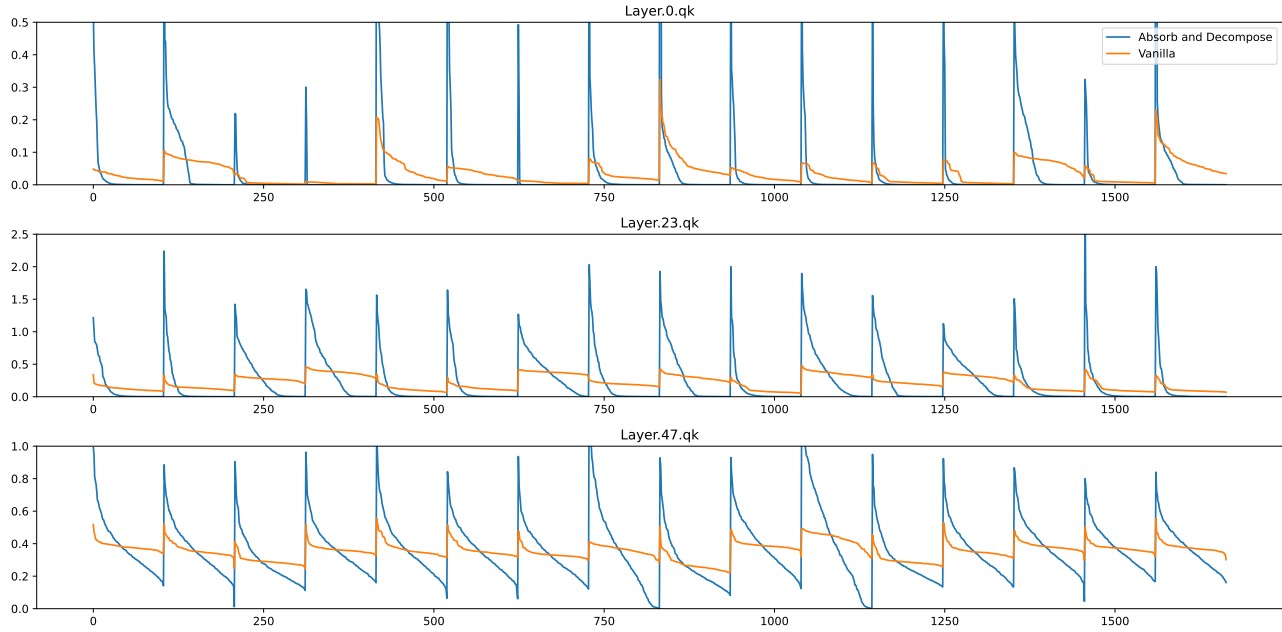

Figure 9. The $L_2$-norm for the 0-th, 15-th, and 31-st attention layers in the ViT-bigG. The blue line represents the results after redundancy removal using the CLOVER method, while the orange line depicts the $L_2$-norm directly computed for each dimension.

