# OpenReview forum: "CLOVER: Cross-Layer Orthogonal Vectors Pruning"
_ICML.cc/2025/Conference — ICML 2025 poster_

### Official Review · Reviewer_AvA1 · 2025-03-14

**Overall Recommendation:** 3

**Summary:**

This paper proposes a method dubbed CLOVER to address the memory-bound in large language models during inference. Specifically, CLOVER performs singular value decomposition (SVD) on the Query-Key and Value-Output parameter matrices in the attention layer, thereby orthogonalizing the vectors within attention heads to reduce linear redundancy. This enables efficient pruning and parameter efficient fine-tuning. Experimental results demonstrate that CLOVER can remove more redundancy than the vanilla pruning method and achieves performance improvements with parameter efficient but full-rank fine-tuning while avoiding the issue of intrusive dimensions.

**Claims And Evidence:**

Yes. The claim in the paper about cross-layer decomposition is mathematically sound and has experimental validation.

**Essential References Not Discussed:**

The paper provides a comprehensive enough discussion of related work.

**Experimental Designs Or Analyses:**

Yes. The experimental design of the paper is generally well-founded. Specifically, the paper validates the effectiveness of CLOVER for pruning on GPT-2-XL, demonstrates its efficiency for PEFT across the LLAMA series models and multiple inference tasks. Furthermore, more fundamental analysis is provided by observing the parameter and feature space rank, projection distribution, spectral variations, etc.
However, some settings in the experiment are insufficient or may be confusing, please refer to Other Strengths And Weaknesses.

**Methods And Evaluation Criteria:**

Yes. The proposed method facilitates the aggregation of the principal components in the parameter space, exposing redundant components. Additionally, it enables full-rank parameter updates, demonstrating practical significance.

**Other Comments Or Suggestions:**

There are some typos such as  in line 115 “without need introduce” and  in line 181 “We presents”.

**Other Strengths And Weaknesses:**

This paper provides a new perspective on discovering redundancy and improving PEFT performance, which demonstrates application value. The writing is clear and fluent, confirming to the fundamental academic writing standards. However, there are still some weaknesses.
1. The paper claims to address the issue of memory-bound during inference. However, the experiments section does not present any memory-related results to validate the effectiveness of the proposed method. Furthermore, the method is not compared with any novel pruning approaches (such as LLM-Pruner[1], SliceGPT[2], FLAP[3], UKMP[4]) in the targeted inference stage. While a study offering a novel perspective may be tolerated even if it does not achieve optimal performance, it should at least present comparative results in the experiments or demonstrate the effectiveness of combining this method with existing SOTA pruning techniques.
2. The figures in the paper are somewhat difficult to interpret. For example, in Figure 4, what values are being ranked in "Top" "Next" and "Bottom"? In Figure 5, what do the horizontal and vertical axes represent?
3. The quantitative validation of CLOVER on pruning is only conducted on GPT-2-XL. Since Figure 2 already demonstrates that the method effectively exposes redundancy across various models, reporting the pruning results on these models as well would better showcase the effectiveness of the proposed method.
[1] Llm-pruner: On the structural pruning of large language models. NeurIPS 2023.
[2] Slicegpt: Compress large language models by deleting rows and columns. ICLR 2024.
[3] Fluctuation-Based Adaptive Structured Pruning for Large Language Models. AAAI 2024.
[4] Unified Knowledge Maintenance Pruning and Progressive Recovery with Weight Recalling for Large Vision-Language Models. AAAI 2025.

**Questions For Authors:**

1. In Figure 2, why is the product of the norms of $W_Q$ and $W_K$ used as the vertical axis? I understand that this is consistent with the CLOVER method, but is this metric also used as an importance criterion in the vanilla pruning method? Based on my understanding of the pruning field, l2-norm-based pruning typically utilizes the norm of either $W_Q$ or $W_K$ individually or their sum, rather than their product. Therefore, I have concerns regarding the correctness of the vanilla pruning method used for comparison.
2. Does the experiment in Section 4.4 maintain the parallelizability of multi-head attention? Specifically, is does the $W_{QK}$ remove the same number of dimensions for each head?

**Relation To Broader Scientific Literature:**

The proposed CLOVER demonstrates applicability in both pruning and PEFT. In the context of pruning, existing structured pruning methods suffer from significant performance degradation. CLOVER effectively reduces redundant dimensions by concentrating the principal components in the parameter space. However, the paper claims to only introduce a novel perspective and does not benchmark its performance against SOTA approaches, which **cannot make me fully convinced**. Regarding PEFT, most existing methods operate within individual layers and incur substantial cost. In contrast, CLOVER employs cross-layer low-rank decomposition to achieve full-rank fine-tuning with reduced cost and mitigates the issue of intrusive dimensions.

**Theoretical Claims:**

Yes. I checked SVD, intrusive dimensions problem, etc. and the paper discusses them correctly.

---

> ### Author Rebuttal · Authors · 2025-04-01
>
> **Q1: Benchmark the effectiveness of the proposed method.**
>
> **A1:** For the pruning experiments, we evaluated the effectiveness of CLOVER using the Wikitext2 eval-dataset with the following settings: batch size of 32, max sequence length of 1600, and tested on an A100-PCIE-40GB GPU with different pruning ratios for GPT-2-XL inference speed. The results are summarized in the table below:
>
> | **Pruning Ratio** | **Vanilla Perplexity** | **CLOVER Perplexity** | **Median Time per Batch (s/batch)** | **Throughput (token/s)** | **Latency (ms/token)** |
> | --- | --- | --- | --- | --- | --- |
> | 0% | 14.78 | 14.78 | 0.0635 | 503.67 | 1.99 |
> | 12.5% | 16.38 | 15.77 | 0.0572 | 559.14 | 1.79 |
> | 25.0% | 17.07 | 16.05 | 0.0504 | 634.83 | 1.58 |
> | 37.5% | 18.14 | 16.48 | 0.0505 | 633.06 | 1.58 |
> | 50.0% | 19.02 | 17.13 | 0.0412 | 777.09 | 1.29 |
> | 62.5% | 21.44 | 18.40 | 0.0463 | 690.89 | 1.45 |
> | 75.0% | 27.22 | 20.99 | 0.0390 | 820.40 | 1.22 |
>
> The results clearly demonstrate that pruning the head dimension significantly accelerates model inference. Notably, CLOVER pruning 50% of the head dimensions outperforms vanilla pruning at 37.5% by generating 144 more tokens per second while maintaining a lower perplexity.
>
> For the PEFT benchmarking experiment, please refer to **A2 for Reviewer DDfo**.
>
> **Q2: Conduct pruning on models other than GPT-2-XL and combine CLOVER with existing SOTA pruning techniques.**
>
> **A2:** Thank you for providing a list of some state-of-the-art pruning research. In Section 4.4, we also applied pruning to Whisper-large-v3 in addition to GPT-2-XL. Due to time constraints, we only compared CLOVER with SliceGPT during the rebuttal period. **In the camera-ready version, we will cite them all and compare CLOVER with those methods.** As mentioned in **A3 for Reviewer jpx5**, we expanded our experiments by using CLOVER pruning on Deepseek and OPT models. Early results indicate that CLOVER achieves a 5% pruning ratio improvement on these models, and we plan to refine these experiments further. Additionally, CLOVER supports pruning in models such as ViT and DiT, and we will continue to compare CLOVER with pruning methods in these domains.
>
> **Q3: L2-norm-based pruning typically utilizes the norm of either W_q or W_k individually, or their sum, rather than their product.**
>
> **A3:** You are correct that many L2-norm-based pruning methods use the norm of either W_q or W_k individually or their sum. However, there are cases where the product of both norms is used. For example:
>
> 1.	When pruning the output of one layer affects the input of the next, considering both layers together may yield better results, and in such cases, the product of the two layers is used [1].
>
> 2.	In the attention layer, the attention score is influenced by both Q and K, which is why their product is considered in pruning [2].
>
> **Q4: Does the experiment in Section 4.4 maintain the parallelizability of multi-head attention? Specifically, does it remove the same number of dimensions for each head?**
>
> **A4:** Yes, all pruning in this paper maintains the parallelizability of multi-head attention. In Section 4.1, the pruning ratio is consistent across all layers, with each attention head having the same dimension. In Section 4.4, to achieve higher pruning rates while maintaining a training-free approach, we allow for different pruning ratios for each layer while ensuring that the number of dimensions removed is the same for each attention head within the same layer. This design ensures that inference can proceed in parallel.
>
> **Q5: Figure annotations and typo corrections.**
>
> Thank you for pointing out these issues. In the camera-ready version, we will revise the figure captions for clarity and make necessary corrections to improve the overall expression.
>
> **Q5.1: In Figure 4, what values are being ranked in “Top,” “Next,” and “Bottom”?**
>
> **A5.1:** As explained in [Lines 375-377, left] and [Lines 362-364, right], after performing SVD on the model, According to the magnitude of the singular values, the singular vectors are denoted as: “Top-256,” “Next-256,” and “Bottom-256.” We found that only 10% of the feature components are projected onto the singular vectors corresponding to the Top-256 singular values.
>
> **Q5.2: In Figure 5, what do the horizontal and vertical axes represent?**
>
> **A5.2:** In Figure 5, the X-axis represents the index of the feature values of $\Delta W$, while the Y-axis represents the magnitude of these feature values.
>
> We hope this clarifies your questions and concerns. We greatly appreciate your effort and look forward to your further feedback on our response.
>
> **References:**
>
> [1] Fluctuation-based Adaptive Structured Pruning for Large Language Models, AAAI 2024.
>
> [2] Round and Round We Go! What makes Rotary Positional Encodings useful?, ICLR 2025.

---

### Official Review · Reviewer_grik · 2025-03-14

**Overall Recommendation:** 3

**Summary:**

The paper addresses the memory increase in large language models due to kv caching and applies SVD to the pairs of KV and QO matrices. After an SVD decomposition, the new representation can be used for efficient pruning or for fine-tuning. The authors include experiments on both these fronts, showing that the proposed method allows for higher pruning ratios for the same performance compared to Vanilla pruning improvements over various PEFT methods.

**Claims And Evidence:**

Yes.

**Essential References Not Discussed:**

I believe that all essential references are included.

**Experimental Designs Or Analyses:**

The experiments are in general sound. An issue is with section 4.4 where the results are not convincing. The more important aspect is the lack of baselines form the pruning literature in Section 4.1 and Table 1.

**Methods And Evaluation Criteria:**

The proposed methods and evaluation criteria make sense for the fine-tuning part. For pruning, the only baseline is vanilla pruning.

**Other Comments Or Suggestions:**

1. Abstract: “these values are reintegrated…count” can be rephrased for clarity
2. Figure 2 y axis can be the singular value itself with x axis the index. The current presentation is confusing.
3. Section 4.4 presents a single example (e.g. Figure 3 adds no value) and it seems like cherry picking. Imo should be moved to appendix or removed altogether.
4. Conclusion: CLVOER instead of CLOVER

**Other Strengths And Weaknesses:**

## Strengths

1. The paper addresses two important issues in the literature
2. The idea to treat the KV and VO matrices as pairs is interesting. The writing on the method is clear and the method itself intuitive. (some other parts are not clear \-\> see weaknesses)
3. The experimental validation contains multiple different settings to test the method’s effectiveness..

## Weaknesses

1. The paper is not clear at some points.
   1. How is section 4.5 a fair comparison given that LoRA is is by design low-rank?
   2. See questions for other instances
2. The hyperparameter selection seems to be not thorough: only one learning rate per method and one seed (?). Another example can be “We adjust the learning rate from 6e-4 to 6e-3 and remove weight decay, while keeping other hyperparameters consistent with the other two methods.”
3. The pruning literature is vast but only vanilla pruning is used for comparison

**Questions For Authors:**

1. What are the “series” and “parallel” baselines?
2. L128 (2nd column): “Instead..overhead”. Is the comparison here fair? $D$ corresponds to all heads but is compared with a single head?
3. What is CLOVER with the cross superscript? It has not been introduced in the manuscript.
4. Can you explain the vast performance difference compared to LoRA and DoRA? Are these methods properly tuned?
5. L262 “Due to the non-linear…matrix”: can you clarify this sentence?
6. Xaxis of Figure 1d?

**Relation To Broader Scientific Literature:**

The paper discusses connections with the PEFT and LLM compression literature and uses standard (foundation) models for evaluation.

**Theoretical Claims:**

There are no theoretical claims.

---

> ### Author Rebuttal · Authors · 2025-04-01
>
> **Q1: Is the comparison in Section 4.5 fair, considering that LoRA is designed for low-rank?**
>
> **A1:** As noted in **A2 for Reviewer DDfo**, CLOVER and LoRA have an identical number of trainable parameters. CLOVER offers slight improvements in both training time and GPU memory consumption when compared to LoRA.The original paper also present comparisons with HiRA, a full-rank update method.
>
> **Q2.1:** Are the experiments conducted with a single learning rate per method and a single seed?
>
> **A2.1:** The results for LoRA and DoRA presented in Table 2 are directly taken from the original DoRA paper, where the hyperparameters are carefully tuned. Similarly, the results for HiRA are cited from its original publication. In contrast, we introduce new experimental results for PiSSA and CLOVER, both of which are optimized with the best learning rates and aligned hyperparameters. Specifically, PiSSA achieves optimal performance at a learning rate of 2e-5, while CLOVER performs best at 1e-4, as shown in the table below:
>
> | **Method** | **Learning Rate** | **Acc** |
> | --- | --- | --- |
> | PiSSA | 1e-4 | 80.1 |
> |  | 5e-5 | 82.9 |
> |  | 3e-5 | 84.1 |
> |  | 2e-5 | **84.5** |
> |  | 1e-5 | 83.6 |
> | CLOVER | 5e-4 | 79.0 |
> |  | 2e-4 | 83.9 |
> |  | 1e-4 | **85.2** |
> |  | 5e-5 | 84.3 |
> |  | 2e-5 | 82.8 |
>
> In the original paper, PiSSA and CLOVER were trained using the default seed (42). Following your suggestion, we tested multiple seeds (40, 41, and 42). We will conduct these experiments on additional models and include the updates in the camera-ready version.
>
> | **Model** | **Method** | **Avg.** |
> | --- | --- | --- |
> | LLaMA-7B | PiSSA | 82.7±0.06 |
> |  | CLOVER | 83.3±0.35 |
> | LLaMA2-7B | PiSSA | 84.3±0.26 |
> |  | CLOVER | 84.9±0.23 |
>
> **Q2.2:** Why was the learning rate of CLOVER† adjusted from 6e-4 to 6e-3?
>
> **A2.2:** Both Vanilla Pruning and CLOVER employ full-parameter fine-tuning and same learning rate, while CLOVER† fine-tunes only singular values. Generally, PEFT converge more slowly than full-parameter methods. To speed up convergence, we increased the learning rate and removed weight decay for CLOVER†.
>
> **Q3: How does CLOVER compare to SOTA methods?**
>
> **A3:** Please refer to **A3 for Reviewer jpx5**.
>
> **Q4: What does the y-axis represent in Figure 2?**
>
> **A4:**  After applying CLOVER, we multiply the square root of the singular values by each singular vector and calculate their L2 norms. This result is equivalent to directly using the singular values. Therefore, we uniform this representation for better comparison.
>
>
> **Q5: Section 4.4 presents a single example, which might seem like cherry-picking.**
>
> **A5:** The example is not cherry-picked; it is an official sample provided with the Whisper-large-v3 model [1]. Due to significant linear redundancies in the attention heads of the Whisper model, CLOVER demonstrates substantial effectiveness. We aimed to highlight this clearly in the paper. We plan to systematically evaluate the Whisper model and, as suggested, move this case study to supplementary materials.
>
> **Q6: What are the “series” and “parallel” baselines?**
>
> **A6:** “Series” and “parallel” are two standard baselines commonly used in parameter-efficient fine-tuning for NLP tasks. The series adapter [2] inserts a trainable module sequentially after the Attention and MLP layers, while the parallel adapter [3] integrates a trainable module parallel to the Attention and MLP layers.
>
> **Q7: Why is D compared with a single head in the experiments?**
>
> **A7:** The comparison is fair because we are comparing the reduced dimensions within each attention head, not the entire set of heads against a single head. This process reduces the dimension of each original attention head from $D \times d$ to $D \times r$, where $D$ is the hidden size, $d$ is the attention head dimension, and $r$ represents the attention head rank, with $r \leq d \ll D$.
>
> **Q8: What is CLOVER†?**
>
> **A8:** Please refer to A2 for Reviewer jpx5.
>
> **Q9: Are LoRA and DoRA properly tuned?**
>
> **A9:** Please refer to A2.1.
>
> **Q10: What does L262 mean by “Due to the non-linear…matrix”?**
>
> **A10:** Matrix merging and decomposition are linear operations, which require linearity to preserve equivariance during orthogonalization. However, RoPE encodings vary depending on token positions, introducing non-linearity. This prevents cross-layer merging and decomposition. To address this, we perform orthogonal decomposition within a single layer rather than across layers when fine-tuning Q-K Vectors with RoPE.
>
> **Q11: What does the x-axis represent in Figure 1d?**
>
> **A11:** It represents the pruning ratio.
>
> Thank you for your thoughtful and constructive feedback, which has significantly improved our paper. Please feel free to reach out if you have any further questions or concerns.
>
> [1] https://huggingface.co/openai/whisper-large-v3
>
> [2] Parameter-Efficient Transfer Learning for NLP, ICML 2019.
>
> [3] Towards a Unified View of Parameter-Efficient Transfer Learning, ICLR 2022.

---

### Official Review · Reviewer_jpx5 · 2025-03-15

**Overall Recommendation:** 3

**Summary:**

The manuscript introduces CLOVER which orthogonalizes the Query, Key, Value, and Output vectors in the attention layers, aiming to reduce the computational overhead and thus guiding pruning and serving for effective fine-tuning. Specifically, it is based on treating pairs of attention layers as low-rank decompositions using Singular Value Decomposition (SVD). CLOVER applies SVD to Query-Key (Q-K) and Value-Output (V-O) pairs, allowing the extracted singular values to guide pruning or act as trainable parameters for efficient fine-tuning. Experiments demonstrate the superior performances compared with other state-of-the-art methods such as LoRA, DoRA and PiSSA in PEFT tasks.

**Claims And Evidence:**

This paper overclaims the contributions of the CLOVER framework to cache optimization during the inference phase. The authors attribute the challenges in the development of existing models to "memory and communication bottlenecks" [L051-054 (Left)]. However, it is unclear how the proposed method reduces memory load in the inference phase.

**Essential References Not Discussed:**

There are not any related works that are essential to understanding the key contributions of this paper but not currently cited in this paper.

**Experimental Designs Or Analyses:**

The authors only compare the performance of CLOVER with Vanilla method in the context of pruning, without conducting comparisons with other state-of-the-art methods.
First, they do not explain the specific setting of “Vanilla” method. Second, it is insufficient to demonstrate the effectiveness of CLOVER by merely comparing with Vanilla method. It is recommended that the authors supplement their findings with recent baselines.

**Methods And Evaluation Criteria:**

The descriptions of CLOVER and CLOVER† are somewhat confusing.
In [L141-143 (Right)], the authors demonstrate the CLOVER fine-tuning as “We freeze the matrices UQK h [:, : r] and VQK h [:, : r], and only fine-tune the singular values SQK h [: r, : r].” Besides, in [L190-193(Right)], the authors clarify the CLOVER† as “In the CLOVER† case, after pruning, S is not immediately merged into the U and V matrices but is used for parameter-efficient fine-tuning, with the merging occurring afterward.”
The two descriptions appear to be the same, which raises questions about what is the difference between CLOVER and CLOVER†.

**Other Comments Or Suggestions:**

The abstract of this paper in submitted PDF is inconsistent with that submitted to the ICML system.

**Other Strengths And Weaknesses:**

Strengths:
1.	Efficient Pruning: From theoretic perspective, the method of matrix compression using SVD is novel and effective. It removes redundant vectors while keeping critical features intact, making pruning more effective.
2.	Parameter Efficient Fine-tuning
a)	From performance perspective, CLOVER surpasses LoRA, DoRA, HiRA, and PiSSA on commonsense tasks, achieving up to 7.6% higher accuracy with equal or fewer trainable parameters.
b)	Compared to other PEFT methods, CLOVER still has some advantages. For example, CLOVER alleviates the intrusive dimensions problem identified in LoRA.

Weaknesses:
See “Claims And Evidence”, “Methods And Evaluation Criteria” and “Experimental Designs Or Analyses”.

**Questions For Authors:**

Please clarify the weaknesses above.

**Relation To Broader Scientific Literature:**

This paper shows a promising method in model pruning and parameter-efficient fine-tuning by orthogonalizing the Query, Key, Value, and Output vectors in the attention layers with SVD.

**Theoretical Claims:**

I have checked the correctness of proofs for theoretical claims in Multi-Head Self-Attention Setup, Cross Layers Merging and Cross Layers Orthogonal Decomposition. All the formulations are correct.

---

> ### Author Rebuttal · Authors · 2025-04-01
>
> We sincerely appreciate your thoughtful and constructive feedback. We have carefully addressed each of your concerns and will incorporate your valuable suggestions into the camera-ready version of the paper.
>
> **Q1: Clarify how the proposed method reduces memory load during inference.**
>
> **A1:** Autoregressive token generation in LLMs involves frequent access to the previous key-value (KV) cache, which can significantly hinder inference speed [1]. CLOVER addresses this issue by reducing the dimensionality of attention heads, thereby reducing the KV cache size and effectively lowering memory usage during inference. As noted in **A1 to Reviewer AvA1**, this reduction in KV cache size contributes to more efficient processing during inference.
>
> **Q2: Clarify the confusing descriptions of CLOVER and CLOVER†.**
>
> **A2:** Thank you for pointing out the confusion between CLOVER and CLOVER†. We acknowledge this naming conflict, especially in pruning experiments where both techniques are used together. To resolve this, we will clearly redefine the terms in the camera-ready version:
>
> •	**CLOVER:** After pruning the less important components based on singular value magnitudes, **only the singular values are fine-tuned**, which are then merged back into singular vectors (this corresponds to the original **CLOVER†**).
>
> •	**CLOVER†:** After pruning the less important components, they are directly merged into singular vectors, followed by **full-parameter fine-tuning** (this corresponds to the original **CLOVER**).
>
> This adjustment will ensure clarity and consistency across both the methodology and experimental sections. In this context, CLOVER† serves as an ablation study, validating the effectiveness of the orthogonal initialization method in comparison to vanilla pruning.
>
> **Q3: Setting of Vanilla Pruning and comparison with SoTA methods.**
>
> Vanilla pruning differs from CLOVER primarily in that it does not perform orthogonalization. Instead, we directly prune the head dimension based on the vector norm.  (Lines 182-183 right, 275-293 left). CLOVER is orthogonal to existing pruning methods and can be effectively combined with techniques like SliceGPT [2]. In response to your suggestion, we compared the combination of CLOVER and SliceGPT with SliceGPT alone on both OPT and Deepseek models, reporting the perplexity results on WikiText2, as shown in the table below:
>
> | **Method** | **Attention Pruning Ratio** | **MLP Pruning Ratio** | **OPT 6.7B** | **Deepseek V2-Lite** |
> | --- | --- | --- | --- | --- |
> | Baseline | 0% | 0% | 10.85 | 6.31 |
> | SliceGPT | 25% | 25% | 11.90 | 8.65 |
> | CLOVER | **30%** | 25% | **11.89** | **8.53** |
>
> The baseline represents the original model with no pruning. CLOVER achieves performance comparable to SliceGPT, despite pruning 30% of attention parameters, compared to SliceGPT’s 25%. This demonstrates the simplicity and effectiveness of CLOVER.
>
> It is important to note that most current pruning techniques for Deepseek primarily focus on reducing the number of MoE experts [3] and token selection [4]. There is limited research on pruning attention heads, attention head dimensions, or latent dimensions. CLOVER facilitates pruning these components in Deepseek’s Multi-Layer Attention (MLA), significantly reducing the KV cache overhead. Additionally, CLOVER utilizes a small number of dominant singular values from the Query-Key matrices to dynamically identify important tokens for full computation, enhancing token selection accuracy, especially in long-context tasks when combined with methods like MoBA [4]. Moreover, by rotating Q-K and V-O pairs in attention heads, CLOVER minimizes outliers in weights and activations [5], which is advantageous for reducing quantization errors. We are excited to continue exploring CLOVER to optimize the efficient deployment of Deepseek models.
>
> **Q4: Address the inconsistency in the abstract between the submitted PDF and the ICML submission system.**
>
> **A4:** Thank you for bringing this to our attention. We will promptly update the abstract in the ICML submission system to ensure that it is consistent with the content in the submitted PDF.
>
> We hope this addresses your comments and concerns effectively. We look forward to your feedback and the opportunity to further improve our work.
>
> **References:**
>
> [1] Keep the Cost Down: A Review on Methods to Optimize LLM’s KV Cache Consumption, COLM 2024.
>
> [2] SliceGPT: Compress Large Language Models by Deleting Rows and Columns, ICLR 2024.
>
> [3] MoE-Pruner: Pruning Mixture-of-Experts Large Language Models Using Hints from Its Router, arXiv preprint.
>
> [4] MoBA: Mixture of Block Attention for Long-Context LLMs, arXiv preprint.
>
> [5] SmoothQuant: Accurate and Efficient Post-Training Quantization for Large Language Models, arXiv preprint.

---

### Official Review · Reviewer_DDfo · 2025-03-15

**Overall Recommendation:** 1

**Summary:**

Decoder - only models face memory issues during inference as the key/value cache grows. This paper introduces CLOVER to address this by treating attention layers as low - rank decompositions. CLOVER applies SVD to the Q−K and V−O pairs in each attention head. The resulting singular values can guide pruning or be used for efficient fine - tuning without increasing the model's parameter count.
Experiments on models like GPT - 2 XL, Whisper - Largev3, and LLaMA - 3 show that CLOVER improves pruning efficiency. For example, it can prune more vectors in GPT - 2 XL with less performance degradation. In fine - tuning, CLOVER outperforms state - of - the - art methods such as LoRA, DoRA, HiRA, and PiSSA on commonsense reasoning tasks. It can achieve full - rank updates and better performance with equal or fewer trainable parameters.
Although CLOVER has limitations when dealing with certain non - linear operations, it can be combined with other pruning methods and has potential for future applications, like being combined with quantization methods.

**Claims And Evidence:**

Yes.

**Essential References Not Discussed:**

Yes.

**Experimental Designs Or Analyses:**

Yes.

**Methods And Evaluation Criteria:**

Yes.

**Other Comments Or Suggestions:**

N.A.

**Other Strengths And Weaknesses:**

**Strengths:**

1. This paper is well organized, motivations and method details can be clearly understood.

2. The proposed method can achieve better accuracy with similar or fewer parameters.

3. The paper provides sufficient technical details and implementation parameters, and fine-tunes tasks for as many domains as possible, e.g., NLP, CV.

**Weaknesses:**

1. The domain of this paper is unclear. From my understanding, the proposed method can only be used for models that contain attention mechanisms (please correct me if I am wrong), not all linear-layer models. Although experiments on stable diffusion are included, this is most likely the attention part between the text encoder and u-net, rather than the core component of SD. So, first of all, should the paper even consider including a claim like, for example, a xxx method for large language models? Second, this paper does not seem to explicitly state that this is a PEFT method, while the baselines are almost all PEFT methods. This is important in my opinion, because pruning and efficient fine-tuning are distinct, at least, they should consider the unique characteristics of training and fine-tuning, respectively. Therefore, in summary, two points are unclear: a. whether it can only be used for LLMs; b. whether CLOVER is a PEFT method.

2. I will use the PEFT one to understand CLOVER. As for the PEFT method, the authors show the performance with similar or fewer parameters, which is certainly correct. However, should also consider to show the GPU memory consumption and running time compared to LoRA. For example, the time to run one epoch with the same batch size and max length.

3. Full rank adaptation may be intuitively reasonable, but, to my knowledge, few papers have rigorously demonstrated this. So I do not recommend authors to make such STRONG claims. Moreover, full rank may also lead to information redundancy. The author can consider whether this is one of the main arguments of CLOVER. If so, please elaborate on it. If not, please hide it.

4. Could the authors please provide a comparison of GPU memory consumption between CLOVER and LoRA? No need for complete fine-tuning, just provide the GPU memory data, for example, you can use the smallest batch size to perform several steps.

**Questions For Authors:**

N.A.

**Relation To Broader Scientific Literature:**

Not all. See the weakness 1.

**Theoretical Claims:**

Yes. Mainly for applications, no proof included.

---

> ### Author Rebuttal · Authors · 2025-04-01
>
> **Q1. Clarification of Domain**
>
> **Q1.1: Whether CLOVER is a PEFT method.**
>
> **A1.1:** CLOVER is a straightforward yet effective re-initialization method that benefits **both pruning and PEFT**. Pruning and PEFT are closely interconnected, as both aim to achieve efficient training and inference under resource constraints. Typically, pruning requires a recovery process, and PEFT methods are commonly applied to minimize training overhead and enhance efficiency, particularly in data-scarce scenarios [1]. While many previous pruning methods primarily focus on the pruning phase and directly apply LoRA in the recovery phase [1, 2, 3, 4], **CLOVER uniquely addresses both tasks simultaneously**. It not only enhances pruning rates (comparing with vanilla pruning and SliceGPT, refer to **A3 for reviewer jpx5**) but also improves PEFT training effectiveness. We believe this dual approach provides valuable insights for efficient model deployment.
>
> **Q1.2: Whether CLOVER can only be used for LLMs.**
>
> **A1.2:** Indeed, our proposed CLOVER method targets on the attention component. But since **attention mechanisms are widely used across many domains beyond LLMs** (such as Vision Transformers (ViT) for image classification, detection, segmentation, DiT for image generation, and Automatic Speech Recognition, as well as Text-to-Speech), CLOVER has potential and value in a wide range of applications. Additionally, in Section 4.4, we provide experimental evidence using the **speech recognition** model Whisper, where CLOVER achieves parameter reductions of 56.01% in the Q-K pair and 36.82% in the V-O pair without the need of further fine-tuning. CLOVER’s flexibility allows it to integrate with other techniques, extending its applicability to other models.
>
> **Q2. GPU Memory Consumption and Runtime Comparison with LoRA**
>
> **A2:** We conducted experiments on the LLaMA-2-7B model using commonsense data, trained for 3 epochs with the following hyperparameters: model_max_length = 1024, per_device_train_batch_size = 2, gradient_accumulation_steps = 2, num_gpus = 4, executed on 4 × A800-80G GPUs, with seed = 42. The comparison of memory consumption and runtime is as follows:
>
> | **Method** | **Trainable Parameters** | **Total Parameters** | **Max Allocated Memory** | **Runtime** |
> | --- | --- | --- | --- | --- |
> | LoRA | 56,098,816 | 6,794,514,432 | 110.84 GB | 2:42:37 |
> | CLOVER | 56,098,816 | 6,794,514,432 | 104.75 GB | 2:22:47 |
>
> the results demonstrate that CLOVER consumes less GPU memory and exhibits shorter training runtime compared to LoRA. We attribute this to CLOVER being applied between two layers, whereas LoRA operates in parallel with the main branch. This enables sequential computation, eliminating the need to retain the input features of the main branch.
>
> **Q3. Rigor in Demonstrating Full-Rank Adaptation’s Advantages and Information Redundancy**
>
> **A3:** In Section 4.5, we emphasize the importance of full-rank adaptation by demonstrating that only 6% of the orthogonal components from randomly selected commonsense samples project onto low-rank vectors, leading to inefficient data utilization. Full-rank adaptation resolves this issue by fully utilizing the data’s orthogonal projection components. Sections 4.6 and 4.7 visualize the matrices fine-tuned by various methods, showing that CLOVER’s effects closely resemble full-parameter fine-tuning, whereas LoRA, constrained to low-rank updates, introduces intrusive dimensions. Several studies [5, 6, 7, 8] have rigorously compared full-rank and low-rank adaptation, consistently concluding that full-rank updates outperform low-rank methods. Regarding the concern about information redundancy, we believe this issue is more closely related to the number of trainable parameters rather than the direction of adaptation itself. Since CLOVER maintains parameter efficiency, it mitigates redundancy effectively while offering greater flexibility and learning capacity.
>
> **Q4. Same as Q2**
>
> We hope that these responses address your concerns regarding CLOVER, and we look forward to further discussing any additional points you may have.
>
> ### Reference:
>
> [1] LLM-Pruner: On the Structural Pruning of Large Language Models, NeurIPS 2023.
>
> [2] SlimGPT: Layer-wise Structured Pruning for Large Language Models, NeurIPS 2024.
>
> [3] Pruner-Zero: Evolving Symbolic Pruning Metric from scratch for LargeLanguage Models, ICML2024.
>
> [4] SliceGPT: Compress Large Language Models by Deleting Rows and Columns, ICLR24.
>
> [5] HiRA: Parameter-Efficient Hadamard High-Rank Adaptation for Large Language Models, ICLR 2025.
>
> [6] GaLore: Memory-Efficient LLM Training by Gradient Low-Rank Projection, ICML 2024.
>
> [7] QuanTA: Efficient High-Rank Fine-Tuning of LLMs with Quantum-Informed Tensor Adaptation, NeurIPS 2024.
>
> [8] Delta-LoRA: Fine-Tuning High-Rank Parameters with the Delta of Low-Rank Matrices, arxiv preprint.

---

> > ### Comment · Reviewer_DDfo · 2025-04-05
> >
> > Thanks for the reply, I still have some questions, which prevent me from improving the score at the moment.
> >
> > **[Q1.1].** The author's explanation of pruning and PEFT is strictly correct. But it seems to have avoided the key issues: 1. Pruning and fine-tuning have different purposes and scenarios. Generally speaking, both are for the pursuit of Efficient AI, that is, training and reasoning efficiency, but the former is mostly for compressing models and improving inference speed (of course, some can also improve training speed). The latter is more oriented to fine-tuning speed and GPU consumption. Furthermore, from an experimental point of view, the authors do not consider any specific pruning baselines. Although I am not very familiar with pruning, there should be no lack of existing advanced pruning methods for LLM or general linear layers (please correct me if I am wrong).
> >
> > For the scenario, the former is for recovery, and the latter is for amplifying and updating downstream related knowledge. These differences are also, in my opinion, the reason why there are few joint studies on the two. This is not a problem of the ability of the method itself (for example, there are also modifications of LoRA to the field of pretraining or pruning), but because it is difficult to fully analyze the motivation and experimental effects of your method in these two fields in the length of a paper. For example, the submitted version of this paper did not even consider introducing any experimental results related to efficiency. Although it was added in the rebuttal, the efficiency results are still incomplete. Compared with performance (acc.) part, the efficiency analysis of this paper can even be ignored.
> >
> > In summary, I do not doubt CLOVER's ability to handle two tasks at the same time. I just don't think this article has clearly described the motivation and experimental results of the method for both scenarios. Therefore, I still think that this article can be written in a clearer and more concise way.
> >
> > **[Q1.2].** I think your argument can convince me, but I firmly believe that technology will continue to improve and develop. I still suggest that you add a limitation section, which will not damage the innovation of the method, but make it clearer.
> >
> > **[Q3].** Thanks to the author for providing literature evidence. However, after careful inspection, I did not find comprehensive and rigorous evidence. For example, in HIRA, only figure 2 conducted an experimental comparison on the commonsense reasoning task. However, there is no empirical evidence in other scenarios and tasks. For example, the experimental results in DyLoRA [R1] are on the GLUE benchmark, and there is no strong correlation between rank and performance. As another example, in NoRM [R2], the authors observed that increasing rank would introduce hallucination noise, which may lead to worse results.
> >
> > Here, I do not doubt that high rank will bring benefits. But my intuition is that it is not certain, because it may mean multiple redundancy. Moreover, objectively speaking, there is no paper that comprehensively and empirically analyzes its impact. So this is obviously a very STRONG conclusion. So I still keep my point of view: if the author insists on this statement, please verify it; otherwise, a simpler way is to delete the part of the statement. If it is deleted, I personally think that the readability of the article has been improved.
> >
> > [R1] DyLoRA: Parameter-Efficient Tuning of Pretrained Models using Dynamic Search-Free Low Rank Adaptation. EACL 2023.
> >
> > [R2] Fine-tuning with Reserved Majority for Noise Reduction. ICLR 2025.
> >
> >
> > **My conclusion:** I appreciate the author's response and efforts. But the comments I mentioned are all to make the article more concise, more focused and enhance the readability of this article. Proposing generalist models that can handle several tasks is certainly a good thing and it is the trend. But if people just claim that the method can do this, I don’t think it will be beneficial to the community. For example, if the Efficient AI community follows this way, there will be very few researchers who will really analyze the essence of pruning and PEFT separately.

---

> > > ### Author Response · Authors · 2025-04-05
> > >
> > > Dear Reviewer DDfo,
> > >
> > > Thank you very much for your sincere suggestions.
> > >
> > > **[A1.1].** Honestly, we had intense discussions about whether to accept this suggestion. Some authors believe that **CLOVER is a novel re-initialization method that benefits pruning, fine-tuning, and even quantization, with potential applications in other areas.** In the next step, we plan to use this cross-head rotation re-initialization method to reduce quantization errors in low-bit attention computation, following SmoothQuant[1] and SpinQuant[2]. However, this would make the paper significantly larger, and for a conference paper, this might not be ideal.
> > >
> > > Therefore, we ultimately decided to **accept your suggestion**. In the camera-ready version, we will restructure the paper, making **pruning** the main focus. We will incorporate the inference efficiency experiments conducted during the rebuttal period [**A1 for Reviewer AvA1**] into the main text. Since pruning includes both compression and fine-tuning phases, the improvements in the fine-tuning phase, relative to methods like LoRA, will be presented as an ablation study. More detailed advantages of CLOVER will be fully explored in a new paper or merged into a journal article.
> > >
> > > It is worth mentioning that during the rebuttal period, we compared CLOVER with the previous state-of-the-art pruning method, **sliceGPT**. Please refer to A3 for Reviewer jpx5, where **CLOVER prunes at a larger ratio while maintaining or lowering perplexity on the OPT and Deepseek models.**
> > >
> > > **[A1.2].** Thank you for acknowledging CLOVER’s generality. In fact, we have already discussed the limitations of CLOVER in the **Conclusion and Limitations Section** of the original paper (lines 408-425). In the camera-ready version, we will ensure that this section is clear and comprehensive.
> > >
> > > **[A3].** Thank you for your suggestion. Indeed, as the rank increases, the fine-tuning performance on downstream tasks improves gradually, as shown in Table 6, Table 15, and Table 18 in LoRA [3], especially when the rank is relatively small. However, this improvement in fitting capability comes at the cost of forgetting more knowledge[4] and introducing more noise[5]. **Both low-rank and full-rank have their advantages and disadvantages, and hardly any paper has definitively proven one to be better than the other.** We will revise the wording in the paper to present full-rank updates as one of the benefits of CLOVER, rather than the sole reason for its superiority over low-rank methods.
> > >
> > > Once again, thank you for your thoughtful reply. We hope our response has addressed your concerns, and we look forward to hearing your further feedback. We remain open to continued discussions on any aspect of the paper.
> > >
> > > [1] SmoothQuant: Accurate and Efficient Post-Training Quantization for Large Language Models, ICML2023.
> > >
> > > [2] SpinQuant: LLM Quantization with Learned Rotations, ICLR 2025.
> > >
> > > [3] LoRA: Low-Rank Adaptation of Large Language Models, ICLR 2022.
> > >
> > > [4] LoRA Learns Less and Forgets Less, TMLR 2024.
> > >
> > > [5] Fine-tuning with Reserved Majority for Noise Reduction.

---

### Decision · Program_Chairs · 2025-05-01

**Decision:**

Accept (poster)

**Comment:**

This paper presents an approach that improves memory efficiency in LLMs by performing an SVD on some matrices in the attention layer, resulting in orthogonal vectors within the attention heads that reduces linear redundancy.  The authors show the resulting model has benefits for token pruning and parameter-efficient finetuning.  The paper was reviewed by four experts, where a majority recommend acceptance.  The lone dissenter noted their primary concern was that the writing changes they recommended are extensive enough to warrant another round of reviews.  The other reviewers echoed the concern with the clarify of the paper in its current form, but have provided a detailed plan on how to address reviewer concerns.  While the ACs finds the writing concern justified, they do not find it warrants overturning the majority of reviewers as the authors plan for addressing these was accepted by reviewers.  The authors are strongly encouraged to consider the reviewer recommendations when revising their paper.